# Unifying Data-Model Sparsity for Class-Imbalanced Graph Representation Learning

## Abstract

To relieve the massive computation cost in the field of deep learning, models with more compact architectures have been proposed for comparable performance. However, it is not only the cumbersome model architectures but also the massiveness of the training data that adds up to the expensive computational burdens. This problem is particularly accentuated in the graph learning field: on one hand, Graph Neural Networks (GNNs) trained upon non-Euclidean graph data often encounter relatively higher time costs, due to their irregular density properties; on the other hand, the natural class-imbalance property accompanied by graphs cannot be alleviated by the massiveness of data, therefore hindering GNNs' ability in generalization. To fully tackle the above issues, *(i) theoretically*, we introduce a hypothesis on to what extent a subset of the training data can approximate the full dataset's learning effectiveness, which is further guaranteed by the gradients' distance between the subset and the full set; *(ii) empirically*, we discover that during the learning process of a GNN, some samples in the training dataset are informative in providing gradients for model parameters update. Moreover, the informative subset evolves as the training process proceeds. We refer to this observation as dynamic data sparsity. We also notice that a pruned sparse contrastive GNN model sometimes *"forgets"* the information provided by the informative subset, reflected in their large loss in magnitudes. Motivated by the above findings, we develop a unified data-model dynamic sparsity framework named **Graph Dec**antation (GraphDec) to address the above challenges. The key idea of GraphDec is to identify the informative subset dynamically during the training process by adopting the sparse graph contrastive learning. Extensive experiments on multiple benchmark datasets demonstrate that GraphDec outperforms state-of-the-art baselines for the class-imbalanced graph/node classification tasks, with respect to classification accuracy and data usage efficiency.

## 1 Introduction

Graph representation learning (GRL) (Kipf & Welling, 2017) has shown remarkable power in dealing with non-Euclidean structure data (e.g., social networks, biochemical molecules, knowledge graphs). Graph neural networks (GNNs) (Kipf & Welling, 2017; Hamilton et al., 2017; Veličković et al., 2018), as the current state-of-the-art of GRL, have become essential in various graph mining applications.

However, in many real-world scenarios, training on graph data often encounters two difficulties: *class imbalance* (Park et al., 2022) and *massive data usage* (Thakoor et al., 2021; Hu et al., 2020). *Firstly*, class imbalance naturally exists in datasets from diverse practical domains, such as bioinformatics and social networks. GNNs are sensitive to this property and can be biased toward the dominant classes. This bias may mislead GNNs' learning process, resulting in underfitting samples that are of real importance to the downstream tasks, and poor test performance at last. *Secondly*, massive data usage requires GNN to perform message-passing over nodes of high degrees bringing about heavy computation burdens. Some calculations are redundant in that not all neighbors are informative regarding learning task-related embeddings. Unlike regular data such as images or texts, the connectivity of irregular graph data invokes random memory access, which further slows down the efficiency of data readout.

Accordingly, recent studies (Chen et al., 2021; Zhao et al., 2021; Park et al., 2022) arise to address the issues of *class imbalance* or *massive data usage* in graph data: *(i)* On one hand, to deal with the *class imbalance* issue in node classification on graphs, GraphSMOTE (Zhao et al., 2021) tries to generate new nodes for the minority classes to balance the training data. Improved upon GraphSMOTE, GraphENS (Park et al., 2022) further proposes a new augmentation method by constructing an ego network to learn the representations of the minority classes. *(ii)* On the other hand, to alleviate the *massive data usage*, (Eden et al., 2018; Chen et al., 2018) explore efficient data sampling policies to reduce the computational cost from the data perspective. From the model improvement perspective, some approaches design the quantization-aware training and low-precision inference method to reduce GNNs' operating costs on data. For example, GLT (Chen et al., 2021) applies the lottery ticket pruning technique (Frankle & Carbin, 2019) to simplify graph data and the GNN model concurrently.

Despite progress made so far, existing methods fail to address the class imbalance and computational burden altogether. Dealing with one may even exacerbate the condition of the other: when tackling the data imbalance, the newly synthetic nodes in GraphSMOTE and GraphENS bring along extra computational burdens for the next-coming training process. While a compact model reduces the computational burden to some extent, we interestingly found that the pruned model easily "forgets" the minorities in class-imbalanced data, reflected in its worse performance than the original model's. To investigate this observation, we

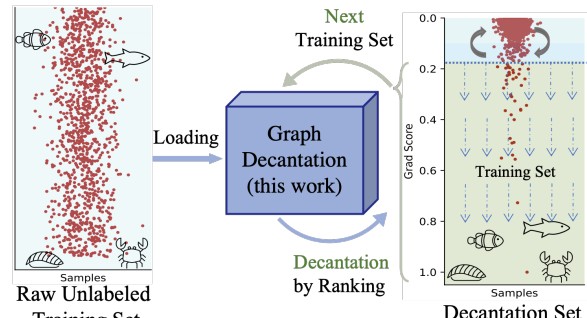

Figure 1: The principle of graph decantation. It decants data samples based on rankings of their gradient scores, and then uses them as the training set in the next epoch.

study how each graph sample affects the GNN training by taking a closer look at the gradients each of them exerts. Specifically, *(i)* in the early phases of training, we identify a small subset that provides the most informative supervisory signals, as measured by the gradient norms' magnitudes (shown in later Figure 5); *(ii)* the informative subset evolves dynamically as the training process proceeds (as depicted in later Figure 3). Both the phenomenons prompt the hypothesize that *the full training set*'s training effectiveness can be approximated, to some extent, by that of the dynamic *subset*. We further show that the effectiveness of the approximation is guaranteed by the distance between the gradients of *the subset* and *the full training set*, as stated in Theorem 1.

Based on the above, we propose a novel method called **Graph Dec**antation (GraphDec) to guide dynamic sparsity training from both the model and data aspects. The principle behind GraphDec is shown in Figure 1. Since the disadvantaged but informative samples tend to bring about higher gradient magnitudes, GraphDec relies on the gradients directed by dynamic sparse graph contrastive learning loss to identify the informative subsets that approximate the full set's training effectiveness. This mechanism not only does not require supervised labels, but also allows for the training of the primary GNN, and the pruning of the sparse one. Specifically, for each epoch, our proposed framework scores samples from the current training set and keep only $k$ most informative samples for the next epoch. Additionally, the framework incorporates a data recycling process, which randomly recycles prior discarded samples (i.e., samples that are considered unimportant in the previous training epochs) by re-involving them in the current training process. As a result, the dynamically updated subset *(i)* supports the sparse GNN to learn relatively unbiased representations and *(ii)* approximates the full training set through the lens of Theorem 1. To summarize, our contributions in this work are:

- We develop a novel framework, Graph Decantation, which leverages dynamic sparse graph contrastive learning on class-imbalanced graph data for efficient data usage. To our best knowledge, this is the first study to explore the dynamic sparsity property for class-imbalanced graphs.

- We introduce cosine annealing to dynamically control the sizes of the sparse GNN model and the graph data subset to smooth the training process. Meanwhile, we introduce data recycling to refresh the current data subset and avoid overfitting.

- Comprehensive experiments on multiple benchmark datasets demonstrate that GraphDec outperforms state-of-the-art methods for both the class-imbalanced graph classification and class-imbalanced node classification tasks. Additional results show that GraphDec dynamically finds an informative subset across the training epochs effectively.

## 2 RELATED WORK

**Graph Contrastive Learning.** Contrastive learning is first established for image tasks and then receives considerable attention in the field of graph representation learning (Chen et al., 2020). Contrastive learning is based on utilizing instance-level identity as supervision and maximizing agreement between positive pairs in hidden space by contrast mode (Velickovic et al., 2019; Hassani & Khasahmadi, 2020; You et al., 2020). Recent research in this area seeks to improve the efficacy of graph contrastive learning by uncovering more difficult views (Xu et al., 2021; You et al., 2021). However, the majority of available approaches utilize a great deal of data. By identifying important subset from the entire dataset, our model avoids this issue.

**Training deep model with sparsity.** Parameter pruning aiming at decreasing computational cost has been a popular topic and many parameter-pruning strategies are proposed to balance the trade-off between model performance and learning efficiency (Deng et al., 2020; Liu et al., 2019). Some of them belong to the static pruning category and deep neural networks are pruned either by neurons (Han et al., 2015b; 2016) or architectures (layer and filter) (He et al., 2017; Dong et al., 2017). In contrast, recent works propose dynamic pruning strategies where different compact subnets will be dynamically activated at each training iteration (Mocanu et al., 2018; Mostafa & Wang, 2019; Raihan & Aamodt, 2020). The other line of computation cost reduction lies in the dataset sparsity (Karnin & Liberty, 2019; Mirzasoleiman et al., 2020; Paul et al., 2021). Recently, the property of sparsity is also used to improve model robustness (Chen et al., 2022; Fu et al., 2021). In this work, we attempt to accomplish dynamic sparsity from both the GNN model and the graph dataset simultaneously.

**Class-imbalanced learning on graphs.** Excepting conventional node re-balanced methods, like reweighting samples (Zhao et al., 2021; Park et al., 2022) and oversampling (Zhao et al., 2021; Park et al., 2022), an early work (Zhou et al., 2018) characterizes rare classes through a curriculum strategy, while other previous works (Shi et al., 2020; Zhao et al., 2021; Park et al., 2022) tackles the class-imbalanced issue by generating synthetic samples to re-balance the dataset. Compared to the node-level task, graph-level re-balancing is under-explored. A recent work (Wang et al., 2021) proposes to utilize neighboring signals to alleviate graph-level class-imbalance. To the best of our knowledge, our proposed GraphDec is the first work to solve the class-imbalanced for both the node-level and graph-level tasks.

## 3 METHODOLOGY

In this section, we first theoretically illustrate our graph sparse subset approximation hypothesis, which guides the design of GraphDec to continuously refine the compact training subset via the dynamic graph contrastive learning method. The presentation is organized by the importance ranking procedure of each sample, refine smoothing, and overfitting regularization. Relevant preliminaries of GNNs, graph contrastive learning, and network pruning are provided in Appendix B.

### 3.1 GRAPH SPARSE SUBSET APPROXIMATION HYPOTHESIS

We first introduce the key notations used in the method. Specifically, we denote the full graph dataset as $\mathcal{G}_F$, the graph data subset used to train the model as $\mathcal{G}_S$, the learning rate as $\alpha$, and the graph learning model parameters as $\theta$ (the optimal model parameters as $\theta*$). Meanwhile, we add a superscript to represent the model's parameters and the graph data subset at epoch $t$, i.e., $\theta^{(t)}$ and $\mathcal{G}_S^{(t)}$. Besides, we use $\mathcal{L}_{\mathcal{G}_S^{(t)};\theta^{(t)}}$ to indicate the loss of model $\theta^{(t)}$ over the graph dataset $\mathcal{G}_S^{(t)}$. Thus, the gradient error at the training epoch $t$ can be computed as $\mathrm{Err}^{(t)} = \left\| \nabla_{\theta^{(t)}} \mathcal{L}_{\mathcal{G}_S^{(t)};\theta^{(t)}} - \nabla_{\theta^{(t)}} \mathcal{L}_{\mathcal{G}_F;\theta^{(t)}} \right\|$. The sparse graph subset approximation hypothesis states that the model effectiveness trained on $\mathcal{G}$ can be approximated by the one trained on $\mathcal{G}_S$. We introduce the hypothesis as follows:

**Theorem 1** *Assume the model's parameters at epoch $t$ satisfies $\left\|\theta^{(t)}\right\|^2 \leqslant d^2$, where $d$ is a constant, and the loss function $\mathcal{L}(\cdot)$ is convex, we can have the following guarantee:*

*If training loss $\mathcal{L}_{\mathcal{G}_S}$ is Lipschitz continuous, $\nabla_{\theta^{(t)}} \mathcal{L}_{\mathcal{G}_S}$ is upper-bounded by $\sigma$, and $\alpha = \frac{d}{\sigma\sqrt{T}}$, then*

$$\min_t(\mathcal{L}_{\mathcal{G}_S^{(t)};\theta^{(t)}} - \mathcal{L}_{\theta*}) \leqslant \frac{d\sigma}{\sqrt{T}} + \sum_{t=1}^{T-1} \frac{d}{T} Err^{(t)}.$$

The detailed proof of Theorem 1 is provided in Appendix A. According to Theorem 1, it is straightforward that we can minimize the gap between the models trained on the full graph dataset and

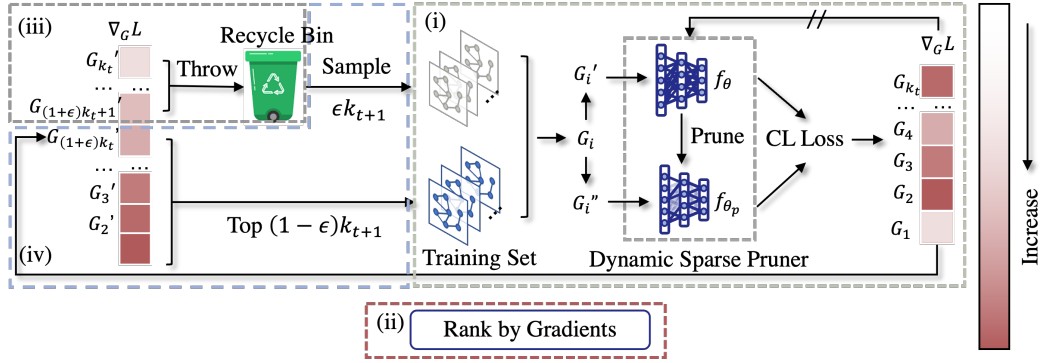

Figure 2: The overall framework of GraphDec: (i) The dynamic sparse graph contrastive learning model computes gradients for graph/node samples; (ii) The input samples are sorted according to their gradients; (iii) Part of the samples with the smallest gradients are thrown into the recycling bin; (iv) Part of the samples with the largest gradients in the current epoch and some sampled randomly from the recycling bin are jointly used as training input in the next epoch.

graph data subset, i.e., $\mathcal{L}_{\mathcal{G}_S;\theta^{(t)}} - \mathcal{L}_{\theta*}$, by reducing the distance between the gradients of the full graph dataset and the graph subset, i.e., $\text{Err}^{(t)}$. In other words, the optimized graph subset $\mathcal{G}_S^{(t)}$ is expected to approximate the gradients of the full graph dataset, and thereby exerts minimal affects on parameters' update. In contrast to GraphDec, data diet (Paul et al., 2021) is designed to identify the most influential data samples $\mathcal{G}_S$ (those with largest gradients during the training phase) only at the early training stage and have them involved in further training processes, while excluding samples from $\bar{\mathcal{G}}_S = \mathcal{G}_F - \mathcal{G}_S$ with smaller gradients (i.e., $\nabla_{\theta^{(t)}}\mathcal{L}_{\mathcal{G}_S^{(t)};\theta^{(t)}} \gg \nabla_{\theta^{(t)}}\mathcal{L}_{\bar{\mathcal{G}}_S^{(t)};\theta^{(t)}}$) eternally. This one-shot selection, however, as we will show in the experiments (Section 4.5), does not always capture the most important samples across all epochs during the training. Specifically, the rankings of elements within a specific $\mathcal{G}_S$ might be relatively static, but those within the full graph dataset, i.e., $\mathcal{G}$, are usually more dynamic, which implies the gradients of the one-shot subset $\nabla_{\theta^{(t)}}\mathcal{L}_{\mathcal{G}_S^{(t)};\theta^{(t)}}$ is unable to constantly approximate that of the full graph dataset $\nabla_{\theta^{(t)}}\mathcal{L}_{\mathcal{G}_F;\theta^{(t)}}$ during training.

## 3.2 GRAPH DECANTATION

Inspired by Theorem 1 and to solve the massive data usage in class-imbalance graphs, we propose GraphDec for achieving competitive performance as well as efficient data usage simultaneously by dynamically filtering out the most influential data subset. The overall framework of GraphDec is illustrated in Figure 2. The training processes are summarized into four steps: (i) First, compute the gradients of the samples in $\mathcal{G}_S^{(t)}$ with respect to the contrastive learning loss; (ii) Normalize the gradients and rank the corresponding graph/node samples in a descending order based on their gradient magnitudes; (iii) Decay the number of samples from $|\mathcal{G}_S^{(t)}|$ to $|\mathcal{G}_S^{(t+1)}|$ with cosine annealing, where we only keep the top $(1-\epsilon)|\mathcal{G}_S^{(t+1)}|$ samples ($\epsilon$ is the exploration rate which controls the ratio of the randomly re-sampled samples from the recycle bin. The rest samples will hold in the recycle bin temporarily; (iv) Finally, randomly re-sample $\epsilon|\mathcal{G}_S^{(t+1)}|$ samples from the recycled bin. The union of these samples and the ones selected in step (iii) will be used for model training in the $(t+1)$-th epoch. Each of the four steps is described in detail in the following content.

**Compute gradients by dynamic sparse graph contrastive learning model.** We adopt the mechanism of dynamic sparse graph contrastive learning in computing the gradients. The reason is two-folded: (a) it scores the graph samples without the supervision of any label; (b) this pruning process is more sensitive in selecting informative samples, verified in Appendix D. We omit the superscript $(t)$ for the dataset and model parameters for simplicity in the explanation of this step. Specifically, given a graph training set $\mathcal{G} = \{G_i\}_{i=1}^N$ as input, for each training sample $G_i$, we randomly generate two augmented graph views, $G_i'$ and $G_i''$, and feed them into the original GCN model $f_\theta(\cdot)$, and the sparse model $f_{\theta_p}(\cdot)$ pruned dynamically by the dynamic sparse pruner, respectively. The gradients are computed based on the outputs of the two GNN branches, directed by the contrastive learning loss signals. To obtain the pruned GNN model, the pruner only keeps neural connections with the top-k largest weight magnitudes. Specifically, the pruned parameters of $l$-th

GNN layer (i.e., $\theta^l$) are selected following the formula below:

$$\theta_p^l = \text{TopK}(\theta^l, k); k = \beta^{(t)} \times |\theta^l|, \tag{1}$$

where $\text{TopK}(\theta^l, k)$ refers to the operation of selecting the top-$k$ largest elements of $\theta^l$, and $\beta^{(t)}$ is the fraction of the remaining neural connections, controlled by the cosine annealing formulated as follows:

$$\beta^{(t)} = \frac{\beta^{(0)}}{2} \left\{ 1 + \cos(\frac{\pi t}{T}) \right\}, t \in [1, T], \tag{2}$$

where $\beta^{(0)}$ is initialized as 1. In addition, we refresh $\theta_p^l$ every few epochs to reactivate neurons based on their gradients, following the formula below:

$$\mathbb{I}_{\theta_g^l} = \text{argTopK}(\nabla_{\theta^l} \mathcal{L}_{D_S;\theta}, k); k = \beta^{(t)} \times |\theta^l|, \tag{3}$$

where $\text{argTopK}$ returns the indices of the top-$k$ largest elements $\mathbb{I}_{\theta_g^l}$ of the corresponding neurons $\theta_g^l$. To further elaborate, we refresh $\theta_p^l$ every few epochs by $\theta_p^l \leftarrow \theta_p^l \cup \theta_g^l$, as the updated pruned parameters to be involved in the next iteration. After we obtain the pruned model, gradients are computed based on the contrastive learning loss between $f_\theta(G_i')$ and $f_{\theta_p}(G_i'')$, which are then saved for the further ranking process.

**Rank graph samples according to their gradients' $L_2$ norms.** In order to find the relative importance of the samples, we rank the samples based on the gradients each of them brings about, saved in the previous training epoch by the last step. Specifically, at each of the $t$-th training epoch, we score each sample by the $L_2$ norm of its gradient:

$$g(G_i) = \left\| \nabla_\theta \mathcal{L} \left( f_\theta(G_i'), f_{\theta_p}(G_i'') \right) \right\|_2, \tag{4}$$

where $\mathcal{L}$ is the popular InfoNCE (Van den Oord et al., 2018) loss in contrastive learning, taking the outputs of the two GNN branches as inputs. Therefore, the gradient is calculated as follows:

$$\nabla_\theta \mathcal{L}(f_\theta(G_i'), f_{\theta_p}(G_i'')) = p_\theta(G') - p_{\theta_p}(G''), \tag{5}$$

where $p_\theta(G_i')$ and $p_{\theta_p}(G_i'')$ are the normalized model's predictions, i.e., $p(\cdot) = S(f(\cdot))$ and $S(\cdot)$ is the softmax function or sigmoid function. The samples are ranked based on the values calculated by Eq. 4.

**Decay the size of $\mathcal{G}_S$ by cosine annealing.** For decreasing the size of the subset, we use cosine annealing when the training process proceeds. As we will show in Figure 3 for the experiments, some graph samples showing low scores of importance at the early training stage may be highly-scored again if given more patience in the later training epochs. Therefore, chunking the size of the sparse subset radically in one shot deprives the chances of the potential samples informing the models at a later stage. To tackle this issue, we employ cosine annealing to gradually decrease the size of the subset:

$$|\mathcal{G}_S^{(t)}| = \frac{|\mathcal{G}|}{2} \left\{ 1 + \cos(\frac{\pi(t)}{T}) \right\}, t \in [1, T]. \tag{6}$$

Note that this process not only automatically decreases the size of $\mathcal{G}_S$ smoothly, but also avoids the manual one-shot selection as in the data diet (Paul et al., 2021).

**Recycle removed graph samples for the next training epoch.** We aim to update the elements in $\mathcal{G}_S^{(t)}$ obtained in the last step. Since current low-scored samples may still have the potential to be highly-scored in the later training epochs, we randomly recycle a proportion of the removed samples and re-involve them in the training process again. Specifically, the exploration rate $\epsilon$ controls the proportion of data that substitutes a number of $\epsilon|\mathcal{G}_S^{t+1}|$ samples with the lowest scores with the same amount of randomly selected samples in $\mathcal{G}_S^{(t+1)}$. At the $t$-th epoch, the update rule is formulated as follows:

$$\mathcal{G}_S^{(t+1)} = \text{TopK}(\mathcal{G}_S^{(t)}, (1-\epsilon)|\mathcal{G}_S^{(t+1)}|) \bigcup \text{SampleK}(\bar{\mathcal{G}_S}^{(t-1)}, \epsilon|\mathcal{G}_S^{(t+1)}|), \tag{7}$$

where $\text{SampleK}(\bar{\mathcal{G}_S}^{(t-1)}, \epsilon|\mathcal{G}_S^{(t+1)}|)$ returns randomly sampled $\epsilon|\mathcal{G}_S^{(t+1)}|$ samples from $\bar{\mathcal{G}_S}^{(t-1)}$, saved in the last epoch. We utilize the compact sparse subset $\mathcal{G}_S^{(t+1)}$ for the training purposes at $(t+1)$-th epoch, and repeat the previous pipelines until T epochs.

# 4 EXPERIMENTS

In this section, we conduct extensive experiments to validate the effectiveness of our proposed model for both the graph and node classification tasks under imbalanced datasets. We also conduct ablation study and informative subset evolution analysis to further prove the effectiveness. Due to space limit, more analysis validating GraphDec's properties and resource cost are provided in Appendix D and E.

## 4.1 EXPERIMENTAL SETUP

**Datasets**. We validate our model on various graph benchmark datasets for the two classification tasks under the class-imbalnced data scenario. For the class-imbalanced graph classification task, we choose the seven validation datasets in G$^2$GNN paper (Wang et al., 2021), i.e., MUTAG, PROTEINS, D&D, NCI1, PTC-MR, DHFR, and REDDIT-B in (Morris et al., 2020). For the class-imbalanced node classification task, we choose the five datasets in the GraphENS paper (Park et al., 2022), i.e., Cora-LT, CiteSeer-LT, PubMed-LT (Sen et al., 2008), Amazon-Photo, and Amazon-Computers. Detailed descriptions of these datasets are provided in the Appendix C.1.

**Baselines**. We compare our model with a variety of baselines methods with different rebalance methods. For class-imbalanced graph classification, we consider three rebalance methods, i.e., vanilla (without re-balancing when training), up-sampling (Wang et al., 2021), and re-weight (Wang et al., 2021). For each rebalance method, we run three baseline methods including GIN (Xu et al., 2019), InfoGraph (Sun et al., 2019), and GraphCL (You et al., 2020). In addition, we adopt two versions of G$^2$GNN (i.e., remove-edge and mask-node) (Wang et al., 2021) for in-depth comparison. For class-imbalanced node classification, we consider nine baseline methods including vanilla, SynFlow (Tanaka et al., 2020), BGRL (Thakoor et al., 2021), GRACE (Zhu et al., 2020), re-weight (Japkowicz & Stephen, 2002), oversampling (Park et al., 2022), cRT (Kang et al., 2020), PC Softmax (Hong et al., 2021), DR-GCN (Shi et al., 2020), GraphSMOTE (Zhao et al., 2021), and GraphENS (Park et al., 2022). We adopt Graph Convolutional Network (GCN) (Kipf & Welling, 2017) as the default architecture for all rebalance methods. Further details about the baselines are illustrated in Appendix C.2.

**Evaluation Metrics**. To evaluate model performance, we choose F1-micro (F1-mi.) and F1-macro (F1-ma.) scores as the metrics for the class-imbalanced graph classification task, and accuracy (Acc.), balanced accuracy (bAcc.), and F1-macro (F1-ma.) score for the node classification task.

**Experimental Settings**. We adopt GCN (Kipf & Welling, 2017) as the GNN backbone of GraphDec for both the tasks. In particular, we concatenate a two-layers GCN and a one-layer fully-connected layer for node classification, and add one extra average pooling operator as the readout layer for graph classification. We follow (Wang et al., 2021) and (Park et al., 2022) varying the imbalance ratios for graph and node classification tasks, respectively. In addition, we take GraphCL (You et al., 2020) as the graph contrastive learning framework, and cosine annealing to dynamically control the sparsity rate in the GNN model and the dataset. The target pruning ratio for the model is set to 0.75, and the one for the dataset is set to 1.0. After the contrastive pre-training, we take the GCN output logits as the input to the Support Vector Machine for fine-tuning. GraphDec is implemented in PyTorch and trained on NVIDIA V100 GPU.

## 4.2 CLASS-IMBALANCED GRAPH CLASSIFICATION PERFORMANCE

The evaluated results for the graph classification task on class-imbalanced graph datasets are reported in Table 1, with the best performance and runner-ups bold and underlined, respectively. From the table, we find that GraphDec outperforms baseline methods on both the metrics across different datasets, while only uses an average of 50% data and 50% model weights per round. Although a slight F1-micro difference has been detected on D&D when comparing GraphDec to the best baseline G$^2$GNN, it is understandable due to the fact that the graphs in D&D are significantly larger than those in other datasets, necessitating specialized designs for graph augmentations (e.g., the average graph size in terms of node number is 284.32 for D&D, but 39.02 and 17.93 for PROTEINS and MUTAG, respectively). However, in the same dataset, G$^2$GNN only achieves 43.93 on F1-macro while GraphDec reaches to 44.01, which complements the 2% difference on F1-micro and further demonstrates GraphDec's ability to learn effectively even on large graph datasets. Specifically, models trained under the vanilla setting perform the worst due to the ignorance of the class-imbalance. Up-sampling strategy improves the performance, but it introduces additional unnecessary data usage by sampling the minorities multiple times. Similarly, re-weight strategy tries to address the class-

Table 1: Class-imbalanced graph classification results. Numbers after each dataset name indicate imbalance ratios of minority to majority categories. Best/second-best results are in bold/underline.

| Rebalance Method | Basis | MUTAG (5:45) | | PROTEINS (30:270) | | D&D (30:270) | | NCI1 (100:900) | | Sparsity (%) | |
|---|---|---|---|---|---|---|---|---|---|---|---|
| | | F1-ma. | F1-mi. | F1-ma. | F1-mi. | F1-ma. | F1-mi. | F1-ma. | F1-mi. | data | model |
| vanilla | GIN | 52.50 | 56.77 | 25.33 | 28.50 | 9.99 | 11.88 | 18.24 | 18.94 | 100 | 100 |
| | InfoGraph | 69.11 | 69.68 | 35.91 | 36.81 | 21.41 | 27.68 | 33.09 | 34.03 | 100 | 100 |
| | GraphCL | 66.82 | 67.77 | 40.86 | 41.24 | 21.02 | 26.80 | 31.02 | 31.62 | 100 | 100 |
| up-sampling | GIN | 78.03 | 78.77 | 65.64 | 71.55 | 41.15 | 70.56 | 59.19 | 71.80 | >100 | 100 |
| | InfoGraph | 78.62 | 79.09 | 62.68 | 66.02 | 41.55 | 71.34 | 53.38 | 62.20 | >100 | 100 |
| | GraphCL | 80.06 | 80.45 | 64.21 | 65.76 | 38.96 | 64.23 | 49.92 | 58.29 | >100 | 100 |
| re-weight | GIN | 77.00 | 77.68 | 54.54 | 55.77 | 28.49 | 40.79 | 36.84 | 39.19 | 100 | 100 |
| | InfoGraph | 80.85 | 81.68 | 65.73 | 69.60 | 41.92 | 72.43 | 53.05 | 62.45 | 100 | 100 |
| | GraphCL | 80.20 | 80.84 | 63.46 | 64.97 | 40.29 | 67.96 | 50.05 | 58.18 | 100 | 100 |
| G²GNN | remove edge | 80.37 | 81.25 | 67.70 | 73.10 | 43.25 | 77.03 | 63.60 | 72.97 | 100 | 100 |
| | mask node | 83.01 | 83.59 | 67.39 | 73.30 | 43.93 | **79.03** | 64.78 | 74.91 | 100 | 100 |
| GraphDec | dynamic sparsity | **85.71** | **85.71** | **68.32** | **75.84** | **44.01** | 77.02 | **65.73** | **76.02** | 50 | 50 |

| Rebalance Method | Basis | PTC-MR (9:81) | | DHFR (12:108) | | REDDIT-B (50:450) | | Avg. Rank | | Sparsity (%) | |
|---|---|---|---|---|---|---|---|---|---|---|---|
| | | F1-ma. | F1-mi. | F1-ma. | F1-mi. | F1-ma. | F1-mi. | F1-ma. | F1-mi. | data | model |
| vanilla | GIN | 17.74 | 20.30 | 35.96 | 49.46 | 33.19 | 36.02 | 12.00 | 12.00 | 100 | 100 |
| | InfoGraph | 25.85 | 26.71 | 50.62 | 56.28 | 57.67 | 67.10 | 11.00 | 11.14 | 100 | 100 |
| | GraphCL | 24.22 | 25.16 | 50.55 | 56.31 | 53.40 | 62.19 | 10.71 | 10.57 | 100 | 100 |
| up-sampling | GIN | 44.78 | 55.43 | 55.96 | 59.39 | 66.71 | 83.00 | 6.00 | 5.43 | >100 | 100 |
| | InfoGraph | 44.29 | 48.91 | 59.49 | 61.62 | 67.01 | 78.68 | 6.00 | 6.00 | >100 | 100 |
| | GraphCL | 45.12 | 53.50 | 60.29 | 61.71 | 62.01 | 75.84 | 6.29 | 6.43 | >100 | 100 |
| re-weight | GIN | 36.96 | 43.09 | 55.16 | 57.78 | 45.17 | 51.92 | 9.86 | 9.86 | 100 | 100 |
| | InfoGraph | 44.09 | 49.17 | 58.67 | 60.24 | 65.79 | 77.35 | 5.43 | 5.29 | 100 | 100 |
| | GraphCL | 44.75 | 52.22 | 60.87 | 61.93 | 62.79 | 76.15 | 6.00 | 6.29 | 100 | 100 |
| G²GNN | remove edge | 46.40 | 56.61 | 61.63 | 63.61 | 68.39 | **86.35** | 2.71 | 2.86 | 100 | 100 |
| | mask node | 46.61 | 56.70 | 59.72 | 61.27 | 67.52 | 85.43 | 2.71 | 2.71 | 100 | 100 |
| GraphDec | dynamic sparsity | **47.07** | **58.15** | **62.25** | **63.61** | **69.70** | **87.00** | 1.00 | 1.14 | 50 | 50 |

imbalanced issue by assigning different weights to different samples. However, it requires the labels for weight calculation and thus may not generalize well when labels are missing. G²GNN, as the best baseline, obtains decent performance by considering the usage of rich supervisory signals from both globally and locally neighboring graphs. Finally, the proposed model, GraphDec, achieves the best performance due to its ability in capturing dynamic data sparsity on from both the model and data perspectives. In addition, we rank the performance of GraphDec with regard to baseline methods on each dataset. GraphDec ranks 1.00 and 1.14 on average, which further demonstrates the superiority of GraphDec. Notice that all existing methods utilize the entire datasets and the model weights while GraphDec only uses half of the data and weights to achieve superior performance.

### 4.3 CLASS-IMBALANCED NODE CLASSIFICATION PERFORMANCE

For the class-imbalanced node classification task, we first evaluate GraphDec on three long-tailed citation graphs (i.e., Cora-LT, CiteSeer-LT, PubMed-LT) and report the results on Table 2. We find that GraphDec obtains the best performance compared to baseline methods for different metrics. GraphSMOTE and GraphENS achieve satisfactory performance by generating virtual nodes to enrich the involvement of the minorities. In comparison, GraphDec does not rely on synthetic virtual nodes to learn balanced representations, thereby avoiding the unnecessary computational costs. Similarly to the class-imbalanced graph classification task in Section 4.2, GraphDec leverages only half of the data and weights to achieve the best performance, whereas all baselines perform worse even with the full dataset and weights. To validate the efficacy of the proposed model on the real-world data, we evaluate GraphDec on naturally class-imbalanced benchmark datasets (i.e., Amazon-Photo and Amazon-Computers). We see that GraphDec has the best performance on both datasets, which demonstrates our model's effectiveness with data sourced from different practical scenes.

### 4.4 ABLATION STUDY

Since GraphDec is a unified learning framework relying on multiple components (steps) to employ dynamic sparsity training from both the model and dataset perspectives, we conduct ablation study to

Table 2: Class-imbalanced node classification results. Best/second-best results are in bold/underline.

| Method | Cora-LT | | | CiteSeer-LT | | | PubMed-LT | | | A.P. ($\rho$ =82) | | A.C. ($\rho$ =244) | | Sparsity (%) | |
|---|---|---|---|---|---|---|---|---|---|---|---|---|---|---|---|
| | Acc. | bAcc. | F1-ma. | Acc. | bAcc. | F1-ma. | Acc. | bAcc. | F1-ma. | (b)Acc. | F1-ma. | (b)Acc. | F1-ma. | data | model |
| vanilla | 73.66 | 62.72 | 63.70 | 53.90 | 47.32 | 43.00 | 70.76 | 57.56 | 51.88 | 82.86 | 78.72 | 68.47 | 64.01 | 100 | 100 |
| SynFlow | 72.98 | 60.62 | 63.29 | 52.85 | 46.23 | 42.19 | 69.63 | 56.75 | 50.99 | 81.57 | 76.93 | 68.10 | 62.97 | 100 | - |
| GRACE | 74.72 | 63.95 | 65.26 | 54.94 | 50.87 | 46.90 | 72.37 | 63.22 | 58.18 | 83.57 | 83.61 | 73.02 | 64.52 | 100 | 100 |
| BGRL | 73.81 | 64.95 | 64.87 | 56.84 | 50.83 | 47.04 | 74.17 | 62.21 | 59.07 | 83.49 | 82.37 | 75.88 | 63.15 | 100 | 100 |
| Re-Weight | 75.20 | 68.79 | 69.27 | 62.56 | 55.80 | 53.74 | 77.44 | 72.80 | 73.66 | 92.94 | 92.95 | 90.04 | 90.11 | 100 | 100 |
| Oversampling | 77.44 | 70.73 | 72.40 | 62.78 | 56.01 | 53.99 | 76.70 | 68.49 | 69.50 | 92.46 | 92.47 | 89.79 | 89.85 | >100 | 100 |
| cRT | 76.54 | 69.26 | 70.95 | 60.60 | 54.05 | 52.36 | 75.10 | 67.52 | 68.08 | 91.24 | 91.17 | 86.02 | 86.00 | 100 | 100 |
| PC Softmax | 76.42 | 71.30 | 71.24 | 65.70 | **61.54** | **61.49** | 76.92 | 75.82 | 74.19 | 93.32 | 93.32 | 86.59 | 86.62 | 100 | 100 |
| DR-GCN | 73.90 | 64.30 | 63.10 | 56.18 | 49.57 | 44.98 | 72.38 | 58.86 | 53.05 | N/A | N/A | N/A | N/A | 100 | 100 |
| GraphSmote | 76.76 | 69.31 | 70.21 | 62.58 | 55.94 | 54.09 | 75.98 | 70.96 | 71.85 | 92.65 | 92.61 | 89.31 | 89.39 | >100 | 100 |
| GraphENS | 77.76 | 72.94 | 73.13 | **66.92** | 60.19 | 58.67 | 78.12 | 74.13 | 74.58 | 93.82 | 93.81 | 91.94 | 91.94 | >100 | 100 |
| GraphDec | **78.29** | **73.94** | **74.25** | 66.90 | 61.56 | 61.85 | **78.20** | **76.05** | **76.32** | **93.85** | **94.02** | **92.19** | **92.16** | 50 | 50 |

Table 3: Ablation study results for both tasks. Four rows of red represent removing four individual components from data sparsity perspective. Four rows of blue represent removing four individual components from model sparsity perspective. Best results are in bold.

| Variant | Class-imbalanced Graph Classification (F1-ma.) | | | | | | | Class-imbalanced Node Classification (Acc.) | | | | |
|---|---|---|---|---|---|---|---|---|---|---|---|---|
| | MUTAG | PROTEINS | D&D | NCI1 | PTC-MR | DHFR | REDDIT-B | Cora-LT | CiteSeer-LT | PubMed-LT | A. Photos | A. Computer |
| GraphDec | **85.71** | **68.32** | **44.01** | **65.73** | **47.07** | 62.25 | **69.70** | **78.29** | **66.90** | **78.20** | **93.85** | **92.19** |
| w/o GS | 80.10 | 63.42 | 36.61 | 61.80 | 42.12 | 48.57 | 61.40 | 68.96 | 60.33 | 56.22 | 73.22 | 67.84 |
| w/o SS | 80.95 | 63.55 | 42.19 | 62.30 | 45.21 | 61.99 | 70.61 | 77.15 | 64.67 | 76.15 | 79.09 | 91.33 |
| w/o CAD | 78.41 | 57.99 | 40.23 | 60.61 | 44.96 | 50.00 | 67.15 | 74.87 | 62.62 | 75.35 | 90.71 | 83.23 |
| w/o RS | 83.21 | 59.32 | 41.65 | 60.51 | 35.21 | 60.99 | 67.61 | 73.27 | 61.32 | 72.02 | 87.11 | 90.38 |
| w/o RM | 44.37 | 40.42 | 38.45 | 34.39 | 32.14 | 43.75 | 64.82 | 70.97 | 54.58 | 70.16 | 79.01 | 65.38 |
| w/o SG | 82.63 | 65.96 | 42.50 | 69.10 | 35.19 | 61.42 | 69.16 | 77.54 | 67.43 | 72.43 | 91.25 | 90.05 |
| w/o CAG | 83.50 | 54.04 | 40.21 | 51.82 | 34.20 | 62.41 | 64.14 | 75.78 | 63.43 | 73.07 | 92.77 | 87.40 |
| w/o RW | 79.25 | 56.33 | 38.34 | 63.00 | 38.00 | 61.53 | 63.16 | 76.46 | 65.36 | 75.54 | 90.54 | 89.10 |
| w/o S.S. | 80.07 | 63.90 | 39.77 | 57.22 | 38.60 | **62.30** | 65.67 | 74.82 | 65.28 | 74.00 | 86.14 | 86.40 |

prove the validity of each component. Specifically, GraphDec relies on four components to address data sparsity and imbalance, including pruning samples by ranking gradients (GS), training with sparse dataset (SS), using cosine annealing to reduce dataset size (CAD), and recycling removed samples (RS), and the other four to address model sparsity and data imbalance, including pruning weights by ranking magnitudes (RM), using sparse GNN (SG), using cosine annealing to progressively reduce sparse GNN's size (CAG), and reactivate removed weights (RW). In addition, GraphDec employs self-supervision to calculate the gradient score. The details of model variants are provided in Appendix C.3. We analyze the contributions of different components by removing each of them independently. Experiments for both tasks are conducted comprehensively for effective inspection. The results are shown in Table 3.

From the table, we find that the performance drops after removing any component, demonstrating the effectiveness of each component. In general, both mechanisms for addressing data and model sparsity contribute significantly to the overall performance, demonstrating the necessity of these two mechanisms in solving sparsity problem. Self-supervision contributes similarly to the dynamic sparsity mechanisms, in that it enables the identification of informative data samples without label supervision. In the dataset dynamic sparsity mechanism, GS and CAD contribute the most as sparse GNN's discriminability identifies hidden dynamic sparse subsets accurately and efficiently. Regarding the model dynamic sparsity mechanism, removing RM and SG leads to a significant performance drop, which demonstrates that they are the key components in training the dynamic sparse GNN from the full GNN model. In particular, CAG enables the performance stability after the model pruning and helps capture informative samples during decantation by assigning greater gradient norms. Among these variants, the full model GraphDec achieves the best result in most cases. indicating the importance of the combination of the dynamic sparsity mechanisms from the two perspectives, and the self-supervision strategy.

## 4.5 ANALYZING EVOLUTION OF SPARSE SUBSET BY SCORING ALL SAMPLES

To show GraphDec's capability in dynamically identifying informative samples, we show the visualization of sparse subset evolution of data diet and GraphDec on class-imbalanced NCI1 dataset in Figure 3. Specifically, we compute 1000 graph samples with their importance scores. These samples are then ranked according to their scores and marked with sample indexes. From the upper figures

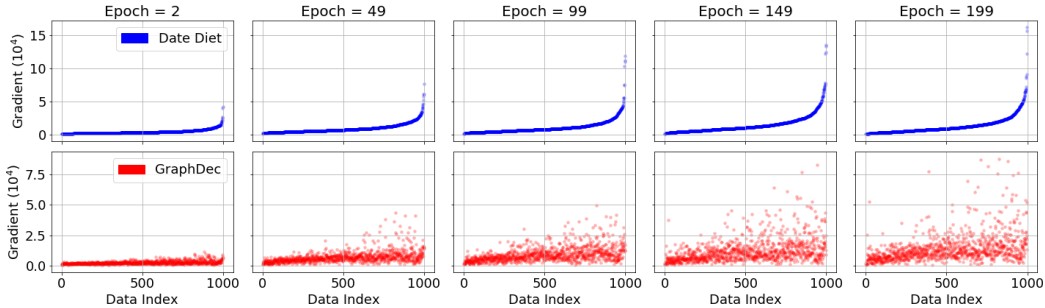

Figure 3: Evolution of data samples' gradients computed by data diet (Paul et al., 2021) (upper figures) and our GraphDec (lower figures) on NCI1 data.

in Figure 3, we find that data diet is unable to accurately identify the dynamic informative nodes. Once a data sample has been removed from the training list due to the low score, the model forever disregards it. However, the fact that a sample is currently unimportant does not imply that it will remain unimportant indefinitely, especially in the early training stage when the model cannot detect the true importance of each sample, resulting in premature elimination of vital nodes. Similarly, if a data sample is considered important at early epochs (i.e., marked with higher sample index), it cannot be removed during subsequent epochs. Therefore, we observe that data diet can only increase the scores of samples within the high index range (i.e., 500–1000), while ignoring samples within the low index range (i.e., < 500). However, GraphDec (Figure 3 (bottom)) can capture the dynamic importance of each sample regardless of the initial importance score. We see that samples with different indexes all have the opportunities to be considered important and therefore be included in the training list. Correspondingly, GraphDec takes into account a broader range of data samples when shrinking the training list, meanwhile maintaining flexibility towards the previous importance scores.

## 5    FINDING INFORMATIVE SAMPLES BY SPARSE GNN

Compared with the full GNN, our dynamic sparse GNN is more sensitive in recognizing informative data samples which can be empirically verified by Figure 4. Our dynamic pruned model assigns larger gradients to the minorities than the majorities during the contrastive training, while the full model generally assigns relatively uniform gradients for both of them. Thus, the proposed dynamically pruned model demonstrates its discriminatory ability on the minority class. This ability in our GraphDec framework is capable of resolving the class-imbalance issue.

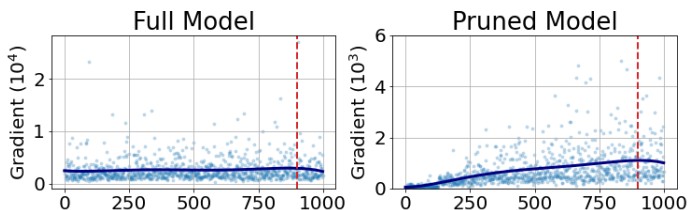

Figure 4: Results of data samples' gradients computed by full GNN model and our dynamic sparse GNN model on NCI1 data. Red dashed line: on the left side, points on the x-axis [0, 900] are majority class; on the right side, points on the x-axis [900, 1000] are minority class.

## 6    CONCLUSION

In this paper, to take up the graph data imbalance challenge, we propose an efficient and effective method named **Graph Dec**antation (GraphDec), by leveraging the dynamic sparse graph contrastive learning to dynamically identified a sparse-but-informative subset for model training, in which the sparse GNN encoder is dynamically sampled from a dense GNN, and its capability of identifying informative samples is used to rank and update the training data in each epoch. Extensive experiments demonstrate that GraphDec outperforms state-of-the-art baseline methods for both node classification and graph classification tasks in the class-imbalanced scenario. The analysis of the sparse informative samples' evolution further explains the superiority of GraphDec in identifying the informative subset among the training periods effectively.

## ETHICS STATEMENT

We do not find that this work is directly related to any ethical risks to society. In general, we would like to see that imbalanced learning algorithms (including this work) are able to perform better on minority groups in real-world applications.

## REPRODUCIBILITY STATEMENT

For the reproducibility of this study, we provide the source code for GraphDec in the supplementary materials. The datasets and other baselines in our experiments are described in Appendix C.1 and C.2.

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

# A  PROOF OF THEOREM 1

We denote the full graph dataset as $\mathcal{G}_F$, the graph data subset used to train the model as $\mathcal{G}_S$, the learning rate as $\alpha$, and the graph learning model parameters as $\theta$ (the optimal model parameters as $\theta*$). Meanwhile, we add a superscript to represent the model's parameters and the graph data subset at epoch $t$, i.e., $\theta^{(t)}$ and $\mathcal{G}_S^{(t)}$. Besides, we use $\mathcal{L}_{\mathcal{G}_S^{(t)};\theta^{(t)}}$ to indicate the loss of model $\theta^{(t)}$ over the graph dataset $\mathcal{G}_S^{(t)}$. Thus, the gradient error at the training epoch $t$ can be computed as $\mathrm{Err}^{(t)} = \left\| \nabla_{\theta^{(t)}} \mathcal{L}_{\mathcal{G}_S^{(t)};\theta^{(t)}} - \nabla_{\theta^{(t)}} \mathcal{L}_{\mathcal{G}_F;\theta^{(t)}} \right\|$. The sparse graph subset approximation hypothesis states that the model effectiveness trained on $\mathcal{G}$ can be approximated by the one trained on $\mathcal{G}_S$. We introduce the hypothesis as follows:

**Theorem 1.** *Assume the model's parameters at epoch $t$ satisfies $\left\| \theta^{(t)} \right\|^2 \leqslant d^2$, where $d$ is a constant, and the loss function $\mathcal{L}(\cdot)$ is convex, we can have the following guarantee:*

*If training loss $\mathcal{L}_{\mathcal{G}_S}$ is Lipschitz continuous, $\nabla_{\theta^{(t)}} \mathcal{L}_{\mathcal{G}_S}$ is upper-bounded by $\sigma$, and $\alpha = \frac{d}{\sigma\sqrt{T}}$, then $\min_t (\mathcal{L}_{\mathcal{G}_S^{(t)};\theta^{(t)}} - \mathcal{L}_{\theta*}) \leqslant \frac{d\sigma}{\sqrt{T}} + \sum_{t=1}^{T-1} \frac{d}{T} Err^{(t)}.$*

**Proof 1** *The gradients of training loss $\mathcal{L}_{\mathcal{G}_S^{(t)};\theta^{(t)}}$ at epoch $t$ are supposed to be $\sigma$-bounded by $\sigma$. According to gradient descent, we have:*

$$\nabla_\theta \mathcal{L}_{\mathcal{G}_S^{(t)};\theta^{(t)}} (\theta^{(t)})^{\mathrm{T}} (\theta^{(t)} - \theta*) = \frac{1}{\alpha^{(t)}} (\theta^{(t)} - \theta^{(t+1)})^{\mathrm{T}} (\theta^{(t)} - \theta*), \tag{8}$$

$$\nabla_\theta \mathcal{L}_{\mathcal{G}_S^{(t)};\theta^{(t)}} (\theta^{(t)})^{\mathrm{T}} (\theta^{(t)} - \theta*) = \frac{1}{2\alpha^{(t)}} \left( \left\| \theta^{(t)} - \theta^{(t+1)} \right\|^2 + \left\| \theta^{(t)} - \theta* \right\|^2 - \left\| \theta^{(t+1)} - \theta* \right\|^2 \right). \tag{9}$$

*Since one update step $\theta^{(t)} - \theta^{(t+1)}$ can be optimized by gradient multiplying with learning rate $\alpha^{(t)} \nabla_\theta \mathcal{L}_{\mathcal{G}_S^{(t)};\theta^{(t)}} (\theta^{(t)})$, we have:*

$$\nabla_\theta \mathcal{L}_{\mathcal{G}_S^{(t)};\theta^{(t)}} (\theta^{(t)})^{\mathrm{T}} (\theta^{(t)} - \theta*) = \frac{1}{2\alpha^{(t)}} \left( \left\| \alpha^{(t)} \nabla_\theta \mathcal{L}_{\mathcal{G}_S^{(t)};\theta^{(t)}} (\theta^{(t)}) \right\|^2 + \left\| \theta^{(t)} - \theta* \right\|^2 - \left\| \theta^{(t+1)} - \theta* \right\|^2 \right). \tag{10}$$

*Since $\nabla_\theta \mathcal{L}_{\mathcal{G}_S^{(t)};\theta^{(t)}} (\theta^{(t)})^{\mathrm{T}} (\theta^{(t)} - \theta*)$ can be represented as follows:*

$$\begin{aligned} \nabla_\theta \mathcal{L}_{\mathcal{G}_S^{(t)};\theta^{(t)}} (\theta^{(t)})^{\mathrm{T}} (\theta^{(t)} - \theta*) = \nabla_\theta \mathcal{L}_{\mathcal{G}_S^{(t)};\theta^{(t)}} (\theta^{(t)})^{\mathrm{T}} (\theta^{(t)} - \theta*) \\ - \nabla_\theta \mathcal{L}_{\mathcal{G}_S^{(t)};\theta^{(t)}}^{\mathrm{T}} (\theta^{(t)} - \theta*) + \nabla_\theta \mathcal{L}_{\mathcal{G}_S^{(t)};\theta^{(t)}}^{\mathrm{T}} (\theta^{(t)} - \theta*), \end{aligned} \tag{11}$$

*then based on the combination of the Equation equation 10 and Equation equation 11, we have:*

$$\begin{aligned} \nabla_\theta \mathcal{L}_{\mathcal{G}_S^{(t)};\theta^{(t)}} (\theta^{(t)})^{\mathrm{T}} (\theta^{(t)} - \theta*) - \nabla_\theta \mathcal{L}_{\mathcal{G}_S^{(t)};\theta^{(t)}}^{\mathrm{T}} (\theta^{(t)} - \theta*) + \nabla_\theta \mathcal{L}_{\mathcal{G}_S^{(t)};\theta^{(t)}}^{\mathrm{T}} (\theta^{(t)} - \theta*) = \\ \frac{1}{2\alpha^{(t)}} \left( \left\| \alpha^{(t)} \nabla_\theta \mathcal{L}_{\mathcal{G}_S^{(t)};\theta^{(t)}} (\theta^{(t)}) \right\|^2 + \left\| \theta^{(t)} - \theta* \right\|^2 - \left\| \theta^{(t+1)} - \theta* \right\|^2 \right) \end{aligned} \tag{12}$$

$$\begin{aligned} \nabla_\theta \mathcal{L}_{\mathcal{G}_S^{(t)};\theta^{(t)}}^{\mathrm{T}} (\theta^{(t)} - \theta*) = \frac{1}{2\alpha^{(t)}} \left( \left\| \alpha^{(t)} \nabla_\theta \mathcal{L}_{\mathcal{G}_S^{(t)};\theta^{(t)}} (\theta^{(t)}) \right\|^2 + \left\| \theta^{(t)} - \theta* \right\|^2 - \left\| \theta^{(t+1)} - \theta* \right\|^2 \right) \\ - \left( \nabla_\theta \mathcal{L}_{\mathcal{G}_S^{(t)};\theta^{(t)}} (\theta^{(t)}) - \nabla_\theta \mathcal{L}_{\mathcal{G}_S^{(t)};\theta^{(t)}} \right)^{\mathrm{T}} (\theta^{(t)} - \theta*). \end{aligned} \tag{13}$$

*We assume learning rate $\alpha^{(t)}, t \in [0, T-1]$ is a constant value, then we have:*

$$\begin{aligned} \sum_{t=0}^{T-1} \nabla_\theta \mathcal{L}_{\mathcal{G}_S^{(t)};\theta^{(t)}}^{\mathrm{T}} (\theta^{(t)} - \theta*) = \frac{1}{2\alpha} \left\| \theta^{(0)} - \theta* \right\|^2 - \left\| \theta^{(t)} - \theta* \right\|^2 + \sum_{t=0}^{T-1} (\frac{1}{2\alpha} \left\| \alpha \nabla_\theta \mathcal{L}_{\mathcal{G}_S^{(t)};\theta^{(t)}} (\theta^{(t)}) \right\|^2) \\ + \sum_{t=0}^{T-1} \left( \left( \nabla_\theta \mathcal{L}_{\mathcal{G}_S^{(t)};\theta^{(t)}} (\theta^{(t)}) - \nabla_\theta \mathcal{L}_{\mathcal{G}_S^{(t)};\theta^{(t)}} \right)^{\mathrm{T}} (\theta^{(t)} - \theta*) \right). \end{aligned}$$

*Since we assume $\left\|\theta^{(t)} - \theta*\right\|^2 \geqslant 0$, then we have:*

$$
\sum_{t=0}^{T-1} \nabla_\theta \mathcal{L}_{\mathcal{G}_S^{(t)};\theta^{(t)}}{}^{\mathrm{T}}(\theta^{(t)} - \theta*) \leqslant \frac{1}{2\alpha}\left\|\theta^{(0)} - \theta*\right\|^2 + \sum_{t=0}^{T-1}(\frac{1}{2\alpha}\left\|\alpha\nabla_\theta \mathcal{L}_{\mathcal{G}_S^{(t)};\theta^{(t)}}(\theta^{(t)})\right\|^2)
$$
$$
+ \sum_{t=0}^{T-1}\left(\left(\nabla_\theta \mathcal{L}_{\mathcal{G}_S^{(t)};\theta^{(t)}}(\theta^{(t)}) - \nabla_\theta \mathcal{L}_{\mathcal{G}_S^{(t)};\theta^{(t)}}\right)^{\mathrm{T}}(\theta^{(t)} - \theta*)\right). \tag{14}
$$

*We assume loss $\mathcal{L}$ is convex and training loss $\mathcal{L}_{\mathcal{G}_S^{(t)};\theta^{(t)}}$ is lipschitz continuous with parameter $\sigma$. Then for convex function $\mathcal{L}(\theta)$, we have $\mathcal{L}_{\mathcal{G}_S^{(t)};\theta^{(t)}} - \mathcal{L}_{\theta*} \leqslant \nabla_\theta \mathcal{L}_{\mathcal{G}_S^{(t)};\theta^{(t)}}{}^{\mathrm{T}}(\theta^{(t)} - \theta*)$. By combining this result with Equation 14, we get:*

$$
\sum_{t=0}^{T-1} \mathcal{L}_{\mathcal{G}_S^{(t)};\theta^{(t)}} - \mathcal{L}_{\theta*} \leqslant \frac{1}{2\alpha}\left\|\theta^{(0)} - \theta*\right\|^2 + \sum_{t=0}^{T-1}(\frac{1}{2\alpha}\left\|\alpha\nabla_\theta \mathcal{L}_{\mathcal{G}_S^{(t)};\theta^{(t)}}(\theta^{(t)})\right\|^2)
$$
$$
+ \sum_{t=0}^{T-1}\left(\left(\nabla_\theta \mathcal{L}_{\mathcal{G}_S^{(t)};\theta^{(t)}}(\theta^{(t)}) - \nabla_\theta \mathcal{L}_{\mathcal{G}_S^{(t)};\theta^{(t)}}\right)^{\mathrm{T}}(\theta^{(t)} - \theta*)\right). \tag{15}
$$

*Since $\left\|\mathcal{L}_{\mathcal{G}_S^{(t)};\theta^{(t)}}(\theta)\right\| \leqslant \sigma$, $\left\|\alpha\nabla_\theta \mathcal{L}_{\mathcal{G}_S^{(t)};\theta^{(t)}}(\theta^{(t)})\right\| \leqslant \sigma$, and we assume $\|\theta - \theta*\| \leqslant d$, then we have:*

$$
\sum_{t=0}^{T-1} \mathcal{L}_{\mathcal{G}_S^{(t)};\theta^{(t)}} - \mathcal{L}_{\theta*} \leqslant \frac{d^2}{2\alpha} + \frac{T\alpha\sigma^2}{2} + \sum_{t=0}^{T-1} d\left(\left\|\nabla_\theta \mathcal{L}_{\mathcal{G}_S^{(t)};\theta^{(t)}}(\theta^{(t)}) - \nabla_\theta \mathcal{L}_{\mathcal{G}_S^{(t)};\theta^{(t)}}\right\|\right), \tag{16}
$$

$$
\frac{1}{T}\sum_{t=0}^{T-1} \mathcal{L}_{\mathcal{G}_S^{(t)};\theta^{(t)}} - \mathcal{L}_{\theta*} \leqslant \frac{d^2}{2\alpha T} + \frac{\alpha\sigma^2}{2} + \sum_{t=0}^{T-1} \frac{d}{T}\left(\left\|\nabla_\theta \mathcal{L}_{\mathcal{G}_S^{(t)};\theta^{(t)}}(\theta^{(t)}) - \nabla_\theta \mathcal{L}_{\mathcal{G}_S^{(t)};\theta^{(t)}}\right\|\right). \tag{17}
$$

*Since $\min\left(\mathcal{L}_{\mathcal{G}_S^{(t)};\theta^{(t)}} - \mathcal{L}_{\theta*}\right) \leqslant \frac{1}{T}\sum_{t=0}^{T-1} \mathcal{L}_{\mathcal{G}_S^{(t)};\theta^{(t)}} - \mathcal{L}_{\theta*}$, based on Equation 17, we have:*

$$
\min\left(\mathcal{L}_{\mathcal{G}_S^{(t)};\theta^{(t)}} - \mathcal{L}_{\theta*}\right) \leqslant \frac{d^2}{2\alpha T} + \frac{\alpha\sigma^2}{2} + \sum_{t=0}^{T-1} \frac{d}{T}\left(\left\|\nabla_\theta \mathcal{L}_{\mathcal{G}_S^{(t)};\theta^{(t)}}(\theta^{(t)}) - \nabla_\theta \mathcal{L}_{\mathcal{G}_S^{(t)};\theta^{(t)}}\right\|\right). \tag{18}
$$

*We set learning rate $\alpha = \frac{d}{\sigma\sqrt{T}}$ and then have:*

$$
\min\left(\mathcal{L}_{\mathcal{G}_S^{(t)};\theta^{(t)}} - \mathcal{L}_{\theta*}\right) \leqslant \frac{d\sigma}{\sqrt{T}} + \sum_{t=0}^{T-1} \frac{d}{T}\left(\left\|\nabla_\theta \mathcal{L}_{\mathcal{G}_S^{(t)};\theta^{(t)}}(\theta^{(t)}) - \nabla_\theta \mathcal{L}_{\mathcal{G}_S^{(t)};\theta^{(t)}}\right\|\right). \tag{19}
$$

## B   PRELIMINARIES: GNNs, GRAPH CONTRASTIVE LEARNING, NETWORK PRUNING

In this work, we denote graph as $G = (V, E, X)$, where $V$ is the set of nodes, $E$ is the set of edges, and $X \in \mathbb{R}^d$ represents the node (and edge) attributes of dimension $d$. In addition, we represent the neighbor set of node $v \in V$ as $N_v$.

**Graph Neural Networks.** GNNs (Wu et al., 2020) learn node representations from the graph structure and node attributes. This process can be formulated as:

$$
h_v^{(l)} = \mathrm{COMBINE}^{(l)}\left(h_v^{(l-1)}, \mathrm{AGGREGATE}^{(l)}\left(\left\{h_u^{(l-1)}, \forall u \in N_v\right\}\right)\right), \tag{20}
$$

where $h_v^{(l)}$ denotes representation of node $v$ at $l$-th GNN layer; $\mathrm{AGGREGATE}(\cdot)$ and $\mathrm{COMBINE}(\cdot)$ are neighbor aggregation and combination functions, respectively; $h_v^{(0)}$ is initialized with node attribute $X_v$. We obtain the output representation of each node after repeating the process in Equation (20) for $L$ rounds. The representation of the whole graph, denoted as $h_G \in \mathbb{R}^d$, can be obtained by using a READOUT function to combine the final node representations learned above:

$$
h_G = \mathrm{READOUT}\left\{h_v^{(L)} \mid \forall v \in V\right\}, \tag{21}
$$

Table 4: Original dataset details for imbalanced graph classification and imbalanced node classification tasks.

| Task | Dataset | # Graphs | # Nodes | # Edges | # Features | # Classes |
|------|---------|----------|---------|---------|------------|-----------|
| Graph | MUTAG | 188 | ~17.93 | ~19.79 | - | 2 |
| | PROTEINS | 1,113 | ~39.06 | ~72.82 | - | 2 |
| | D&D | 1,178 | ~284.32 | ~715.66 | - | 2 |
| | NCI1 | 4,110 | ~29.87 | ~32.30 | - | 2 |
| | PTC-MR | 344 | ~14.29 | ~14.69 | - | 2 |
| | DHFR | 756 | ~42.43 | ~44.54 | - | 2 |
| | REDDIT-B | 2,000 | ~429.63 | ~497.75 | - | 2 |
| Node | Cora | - | 2,485 | 5,069 | 1,433 | 7 |
| | Citeseer | - | 2,110 | 3,668 | 3,703 | 6 |
| | Pubmed | - | 19,717 | 44,324 | 500 | 3 |
| | A-photo | - | 7,650 | 238,162 | 745 | 8 |
| | A-computers | - | 13,381 | 245,778 | 767 | 10 |

where the READOUT function can be any permutation invariant, like summation, averaging, etc.

**Graph Contrastive Learning.** Given a graph dataset $\mathcal{D} = \{G_i\}_{i=1}^{N}$, Graph Contrastive Learning (GCL) methods firstly implement proper transformations on each graph $G_i$ to generate two views $G_i'$ and $G_i''$. The goal of GCL is to map samples within positive pairs closer in the hidden space, while those of the negative pairs are further. GCL methods are usually optimized by a contrastive loss. Taking the most popular InfoNCE loss (Oord et al., 2018) as an example, the contrastive loss is defined as:

$$\mathcal{L}_{CL}(G_i', G_i'') = -\log \frac{\exp\left(\text{sim}\left(\mathbf{z}_{i,1}, \mathbf{z}_{i,2}\right)\right)}{\sum_{j=1, j\neq i}^{N} \exp\left(\text{sim}\left(\mathbf{z}_{i,1}, \mathbf{z}_{j,2}\right)\right)}, \quad (22)$$

where $\mathbf{z}_{i,1} = f_\theta\left(G_i'\right)$, $\mathbf{z}_{i,2} = f_\theta\left(G_i''\right)$, and $\text{sim}$ denotes the similarity function.

**Network Pruning.** Given an over-parameterized deep neural network $f_\theta(\cdot)$ with weights $\theta$, the network pruning is usually performed layer-by-layer. The pruning process of the $l_{th}$ layer in $f_\theta(\cdot)$ can be formulated as follows:

$$\theta_{pruned}^{l_{th}} = \text{TopK}(\theta^{l_{th}}, k), k = \alpha \times |\theta^{l_{th}}|, \quad (23)$$

where $\theta^{l_{th}}$ is the parameters in the $l_{th}$ layer of $f_\theta(\cdot)$ and $\text{TopK}(\cdot, k)$ refers to the operation to choose the top-$k$ largest elements of $\theta^{l_{th}}$. We use a pre-defined sparse rate $\alpha$ to control the fraction of parameters kept in the pruned network $\theta_{pruned}^{l_{th}}$. Finally, only the top $k = \alpha \times |\theta^{l_{th}}|$ largest weights will be kept in the pruned layer. The pruning process will be implemented iteratively to prune the parameters in each layer of deep neural network (Han et al., 2015a).

## C EXPERIMENTAL DETAILS

### C.1 DATASETS DETAILS

In this work, seven graph classification datasets and five node classification datasets are used to evaluate the effectiveness of our proposed model, we provided their detailed statistics in Table 4. For graph classification datasets, we follow the imbalance setting of (Wang et al., 2021) to set the train-validation split as 25%/25% and change the imbalance ratio from 5:5 (balanced) to 1:9 (imbalanced). The rest of the dataset is used as the test set. The specified imbalance ratio of each dataset is clarified after its name in Table 5. For node classification datasets, we follow (Sen et al., 2008) to set the imbalance ratio of Cora, CiteSeer and PubMed as 10. Besides, the setting of Amazon-Photo and Amazon-Computers are borrowed from (Park et al., 2022), where the imbalance ratio $\rho$ is set as 82 and 244, respectively.

### C.2 BASELINE DETAILS

We compare our model with a variety of baseline methods using different rebalance methods:

I. For **imbalanced graph classification** (Wang et al., 2021), four models are included as baselines in our work, we list these baselines as follow:

(1) **GIN** (Xu et al., 2019), a popular supervised GNN backbone for graph tasks due to its powerful expressiveness on graph structure;

(2) **InfoGraph** (Sun et al., 2019), an unsupervised graph learning framework by maximizing the mutual information between the whole graph and its local topology of different levels;

(3) **GraphCL** (You et al., 2020), learning unsupervised graph representations via maximizing the mutual information between the original graph and corresponding augmented views;

(4) **G$^2$GNN** (Wang et al., 2021), a re-balanced GNN proposed to utilize additional supervisory signals from both neighboring graphs and graphs themselves to alleviate the imbalance issue of graph.

II. For **imbalanced node classification**, we consider nine baseline methods in our work, including

(1) **vanilla**, denoting that we train GCN normally without any extra rebalancing tricks;

(2) **re-weight** (Japkowicz & Stephen, 2002), denoting we use cost-sensitive loss and re-weight the penalty of nodes in different classes;

(3) **oversampling** (Park et al., 2022), denoting that we sample nodes of each class to make the data's number of each class reach the maximum number of corresponding class's data;

(4) **cRT** (Kang et al., 2020), a post-hoc correction method for decoupling output representations;

(5) **PC Softmax** (Hong et al., 2021), a post-hoc correction method for decoupling output representations, too;

(6) **DR-GCN** (Shi et al., 2020), building virtual minority nodes and forces their features to be close to the neighbors of a source minority node;

(7) **GraphSMOTE** (Zhao et al., 2021), a pre-processing method that focuses on the input data and investigates the possibility of re-creating new nodes with minority features to balance the training data.

(8) **GraphENS** (Park et al., 2022), proposing a new augmentation method to construct an ego network from all nodes for learning minority representation.

(9) **SynFlow** (Tanaka et al., 2020), a one-shot model pruning method with less reliance on data.

(10) **BGRL** (Thakoor et al., 2021), a graph contrastive learning method using only simple augmentations and avoids the requirements for contrasting with negative examples, and thus makes itself scalable.

(11) **GRACE** (Zhu et al., 2020), a graph contrastive learning method generating two views by corrupting a graph and learning node embeddings by minimizing the distance of node embeddings in these two views.

We use Graph Convolutional Network (GCN) (Kipf & Welling, 2017) as the default architecture for all rebalance methods.

## C.3    Details of GraphDec Variants

The details of model variants are provided as follows:

I. Specifically, GraphDec contains four components to address data sparsity and imbalance: (1) **GS** is sampling informative subset data according to ranking gradients; (2) **SS** is training model with the sparse dataset, correspondingly; (3) **CAD** is using cosine annealing to reduce dataset size; (4) **RS** is recycling removed samples, correspondingly. To investigate their corresponding effectiveness, we remove them correspondingly as:

(1) **w/o GS** is that we randomly sample subset from the full set;

(2) **w/o SS** is that we train GNN with the full set;

(3) **w/o CAD** is that we directly reduce dataset size to target dataset size and it is same as data diet;

Table 5: Imbalanced graph classification results. The numbers after each dataset name indicate the imbalance ratios of minority to majority categories. We report the macro F1-score and micro F1-score with the standard errors as Results are reported as $mean \pm std$ for 3 repetitions on each dataset. We bold the best performance.

| Rebalance Method | Basis | MUTAG (5:45) | | PROTEINS (30:270) | | D&D (30:270) | | NCI1 (100:900) | |
|---|---|---|---|---|---|---|---|---|---|
| | | F1-ma. | F1-mi. | F1-ma. | F1-mi. | F1-ma. | F1-mi. | F1-ma. | F1-mi. |
| vanilla | GIN (Xu et al., 2019) | 52.50 ± 18.70 | 56.77 ± 14.14 | 25.33 ± 7.53 | 28.50 ± 5.82 | 9.99 ± 7.44 | 11.88 ± 9.49 | 18.24 ± 7.58 | 18.94 ± 7.12 |
| | InfoGraph (Sun et al., 2019) | 69.11 ± 9.03 | 69.68 ± 7.77 | 35.91 ± 7.58 | 36.81 ± 6.51 | 21.41 ± 4.51 | 27.68 ± 7.52 | 33.09 ± 3.30 | 34.03 ± 3.68 |
| | GraphCL (You et al., 2020) | 66.82 ± 11.56 | 67.77 ± 9.78 | 40.86 ± 6.94 | 41.24 ± 6.38 | 21.02 ± 3.05 | 26.80 ± 4.95 | 31.02 ± 2.69 | 31.62 ± 3.05 |
| up-sampling | GIN (Xu et al., 2019) | 78.03 ± 7.62 | 78.77 ± 7.67 | 65.64 ± 2.67 | 71.55 ± 3.19 | 41.15 ± 3.74 | 70.56 ± 10.28 | 59.19 ± 4.39 | 71.80 ± 7.02 |
| | InfoGraph (Sun et al., 2019) | 78.62 ± 6.84 | 79.09 ± 6.86 | 62.68 ± 2.70 | 66.02 ± 3.18 | 41.55 ± 2.32 | 71.34 ± 6.76 | 53.38 ± 1.88 | 62.20 ± 2.63 |
| | GraphCL (You et al., 2020) | 80.06 ± 7.79 | 80.45 ± 7.86 | 64.21 ± 2.53 | 65.76 ± 2.61 | 38.96 ± 3.01 | 64.23 ± 8.10 | 49.92 ± 2.15 | 58.29 ± 3.30 |
| re-weight | GIN (Xu et al., 2019) | 77.00 ± 9.59 | 77.68 ± 9.30 | 54.54 ± 6.29 | 55.77 ± 7.11 | 28.49 ± 5.92 | 40.79 ± 11.84 | 36.84 ± 8.46 | 39.19 ± 10.05 |
| | InfoGraph (Sun et al., 2019) | 80.85 ± 7.75 | 81.68 ± 7.83 | 65.73 ± 3.10 | 69.60 ± 3.68 | 41.92 ± 2.28 | 72.43 ± 6.63 | 53.05 ± 1.12 | 62.45 ± 1.89 |
| | GraphCL (You et al., 2020) | 80.20 ± 7.27 | 80.84 ± 7.43 | 63.46 ± 2.42 | 64.97 ± 2.41 | 40.29 ± 3.31 | 67.96 ± 8.98 | 50.05 ± 2.09 | 58.18 ± 3.08 |
| G²GNN (Wang et al., 2021) | remove edge | 80.37 ± 6.73 | 81.25 ± 6.87 | 67.70 ± 2.96 | 73.10 ± 4.05 | 43.25 ± 3.91 | 77.03 ± 9.98 | 63.60 ± 1.57 | 72.97 ± 1.81 |
| | mask node | 83.01 ± 7.01 | 83.59 ± 7.14 | 67.39 ± 2.99 | 73.30 ± 4.19 | 43.93 ± 3.46 | 79.03 ± 10.78 | 64.78 ± 2.86 | 74.91 ± 2.14 |
| GraphDec | dynamic sparsity | **85.71±10.20** | **85.71±11.10** | **68.31±4.23** | **75.84±6.80** | **44.01±5.01** | 77.02±6.26 | **65.73±4.7** | **76.02±6.27** |

| Rebalance Method | Basis | PTC-MR (9:81) | | DHFR (12:108) | | REDDIT-B (50:450) | |
|---|---|---|---|---|---|---|---|
| | | F1-ma. | F1-mi. | F1-ma. | F1-mi. | F1-ma. | F1-mi. |
| vanilla | GIN (Xu et al., 2019) | 17.74 ± 6.49 | 20.30 ± 6.06 | 35.96 ± 8.87 | 49.46 ± 4.90 | 33.19 ± 14.26 | 36.02 ± 17.38 |
| | InfoGraph (Sun et al., 2019) | 25.85 ± 6.14 | 26.71 ± 6.50 | 50.62 ± 8.33 | 56.28 ± 4.58 | 57.67 ± 3.80 | 67.10 ± 4.91 |
| | GraphCL (You et al., 2020) | 24.22 ± 6.21 | 25.16 ± 5.25 | 50.55 ± 10.01 | 56.31 ± 6.12 | 53.40 ± 4.06 | 62.19 ± 5.68 |
| up-sampling | GIN (Xu et al., 2019) | 44.78 ± 8.01 | 55.43 ± 14.25 | 55.96 ± 10.06 | 59.39 ± 6.52 | 66.71 ± 3.92 | 83.00 ± 5.18 |
| | InfoGraph (Sun et al., 2019) | 44.29 ± 4.69 | 48.91 ± 7.49 | 59.49 ± 5.20 | 61.62 ± 4.18 | 67.01 ± 3.34 | 78.68 ± 3.71 |
| | GraphCL (You et al., 2020) | 45.12 ± 7.33 | 53.50 ± 13.31 | 60.29 ± 9.04 | 61.71 ± 6.75 | 62.01 ± 3.97 | 75.84 ± 3.98 |
| re-weight | GIN (Xu et al., 2019) | 36.96 ± 14.08 | 43.09 ± 20.01 | 55.16 ± 9.47 | 57.78 ± 6.69 | 45.17 ± 8.46 | 51.92 ± 12.29 |
| | InfoGraph (Sun et al., 2019) | 44.09 ± 5.62 | 49.17 ± 8.78 | 58.67 ± 5.82 | 60.24 ± 4.80 | 65.79 ± 3.38 | 77.35 ± 3.96 |
| | GraphCL (You et al., 2020) | 44.75 ± 7.62 | 52.22 ± 13.24 | 60.87 ± 6.33 | 61.93 ± 5.15 | 62.79 ± 6.93 | 76.15 ± 9.15 |
| G²GNN (Wang et al., 2021) | remove edge | 46.40 ± 7.73 | 56.61 ± 13.72 | 61.63 ± 10.02 | 63.61 ± 6.05 | 68.39 ± 2.97 | 86.35 ± 2.27 |
| | mask node | 46.61 ± 8.27 | 56.70 ± 14.81 | 59.72 ± 6.83 | 61.27 ± 5.40 | 67.52 ± 2.60 | 85.43 ± 1.80 |
| GraphDec | dynamic sparsity | **47.07±8.22** | **58.15±10.24** | **62.25±9.54** | 63.61±7.10 | **69.70±7.20** | **87.00±9.36** |

Table 6: Imbalanced node classification results. We report the accuracy, balanced accuracy, and macro F1-score with the standard errors as $mean \pm std$ for 3 repetitions on each dataset (Due to time limitation, we will update GRACE, BGRL, and SynFlow's results with standard deviations in next manuscript). We bold the best performance.

| Method | Cora-LT | | | CiteSeer-LT | | | PubMed-LT | | | A.P. ($\rho$ =82) | | A.C. ($\rho$ =244) | |
|---|---|---|---|---|---|---|---|---|---|---|---|---|---|
| | Acc. | bAcc. | F1-ma. | Acc. | bAcc. | F1-ma. | Acc. | bAcc. | F1-ma. | (b)Acc. | F1-ma. | (b)Acc. | F1-ma. |
| vanilla | 73.66±0.28 | 62.72±0.39 | 63.70±0.43 | 53.90±0.70 | 47.32±0.61 | 43.00±0.70 | 70.76±0.74 | 57.56±0.59 | 51.88±0.53 | 82.86±0.30 | 78.72±0.52 | 68.47±2.19 | 64.01±3.18 |
| SynFlow (Tanaka et al., 2020) | 72.98 | 60.62 | 63.29 | 52.85 | 46.23 | 42.19 | 69.63 | 56.75 | 50.99 | 81.57 | 76.93 | 68.10 | 62.97 |
| GRACE (Zhu et al., 2020) | 74.72 | 63.95 | 65.26 | 54.94 | 50.87 | 46.90 | 72.37 | 63.22 | 58.18 | 83.57 | 83.61 | 73.02 | 64.52 |
| BGRL (Thakoor et al., 2021) | 73.81 | 64.95 | 64.87 | 56.84 | 50.83 | 47.04 | 74.17 | 62.21 | 59.07 | 83.49 | 82.37 | 75.88 | 63.15 |
| Re-Weight (Park et al., 2022) | 75.20±0.19 | 68.79±0.18 | 69.27±0.26 | 62.56±0.32 | 55.80±0.28 | 53.74±0.28 | 77.44±0.21 | 72.80±0.38 | 73.66±0.27 | 92.94±0.13 | 92.95±0.13 | 90.04±0.29 | 90.11±0.28 |
| Oversampling (Park et al., 2022) | 77.44±0.09 | 70.73±0.10 | 72.40±0.11 | 62.78±0.37 | 56.01±0.35 | 53.99±0.37 | 76.70±0.48 | 68.49±0.28 | 69.50±0.38 | 92.46±0.47 | 92.47±0.48 | 89.79±0.16 | 89.85±0.17 |
| cRT (Kang et al., 2020) | 76.54±0.22 | 69.26±0.48 | 70.95±0.50 | 60.60±0.25 | 54.05±0.22 | 52.36±0.22 | 75.10±0.23 | 67.52±0.72 | 68.08±0.85 | 91.24±0.28 | 91.17±0.29 | 86.02±0.55 | 86.00±0.56 |
| PC Softmax (Hong et al., 2021) | 76.42±0.34 | 71.30±0.45 | 71.24±0.52 | 65.70±0.42 | 61.54±0.45 | 61.49±0.49 | 76.92±0.26 | 75.82±0.25 | 74.19±0.25 | 93.32±0.25 | 93.32±0.25 | 86.59±0.92 | 86.62±0.91 |
| DR-GCN (Shi et al., 2020) | 73.90±0.29 | 64.30±0.39 | 63.10±0.57 | 56.18±1.10 | 49.57±1.08 | 44.98±1.29 | 72.38±0.19 | 58.86±0.15 | 53.05±0.13 | N/A | N/A | N/A | N/A |
| GraphSmote (Zhao et al., 2021) | 76.76±0.31 | 69.31±0.37 | 70.21±0.64 | 62.58±0.30 | 55.94±0.34 | 54.09±0.37 | 75.98±0.22 | 70.96±0.36 | 71.85±0.32 | 92.65±0.31 | 92.61±0.32 | 89.31±0.34 | 89.39±0.35 |
| GraphENS (Park et al., 2022) | 77.76±0.09 | 72.94±0.15 | 73.13±0.11 | 66.92±0.21 | 60.19±0.21 | 58.67±0.25 | 78.12±0.06 | 74.13±0.22 | 74.58±0.13 | 93.82±0.13 | 93.81±0.12 | 91.94±0.17 | 91.94±0.17 |
| GraphDec | **78.29±0.40** | **73.94±0.67** | **74.25±0.83** | **66.90±0.65** | **61.56±0.72** | **61.85±0.96** | **78.20±0.45** | **76.05±0.66** | **76.32±0.66** | **93.85±0.72** | **94.02±0.67** | **92.19±0.73** | **92.16±0.75** |

(4) **w/o RS** is not recycling any removed samples.

II. Another four components to address model sparsity and data imbalance: (1) **RM** samples model weights according to ranking magnitudes; (2) **SG** is using sparse GNN, correspondingly; (3) **CAG** is using cosine annealing to progressively reduce sparse GNN's size; (4) **RW** is reactivating removed weights. To investigate their effectiveness, we remove them correspondingly as:

(1) **w/o RM** is that we randomly sample activated weights from full GNN model;

(2) **w/o SG** is that we train full GNN during forward and backward;

(3) **w/o CAG** is that we directly reduce the model size to target sparsity rate;

(4) **w/o RW** is not reactivating any removed weights during sparse training.

## C.4 FULL RESULTS WITH ERROR BARS

We provide the F1-macro and F1-micro scores along with their standard deviation for our model and other baselines across both graph classification and node classification tasks in Table 5 and Table 6. We report their results as $mean \pm std$ for 3 repetitions on each metric for each dataset.

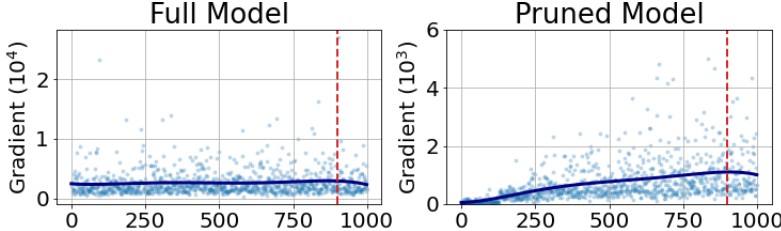

Figure 5: Results of data samples' gradients computed by full GNN model and our dynamic sparse GNN model on NCI1 data. Red dashed line: on the left side, points on the x-axis [0, 900] are majority class; on the right side, points on the x-axis [900, 1000] are minority class.

## D  FINDING INFORMATIVE SAMPLES BY SPARSE GNN

Compared with the full GNN model, our dynamic sparse GNN model is more sensitive in recognizing informative data samples which can be empirically verified by Figure 5. As we can see in the figure, our dynamic pruned model assigns larger gradients to the minorities than the majorities during the contrastive training, while the full model generally assigns relatively uniform gradients for both of them. Thus, the proposed dynamically pruned model demonstrates its discriminatory ability on the minority class.

## E  RESOURCE COST

To evaluate the proposed GraphDec's computational cost on a wide range of datasets, results in Table 7 that include three different class-imbalanced node classification datasets (PubMed-LT, Cora-LT, CiteSeer-LT), three different class-imbalanced graph classification datasets (MUTAG, PROTEINS, PTC_MR), and four baselines (vanilla GCN, re-weight, re(/over)-sample, GraphCL). We run 200 epochs for each method to measure their computational time (second) for training. On NVIDIA GeForce RTX 3090 GPU device, we obtain the running time as reported in Table 7. All models are implemented in PyTorch Geometric (Fey & Lenssen, 2019).

Table 7: Computational time comparisons.

| Model | Method | PubMed-LT | Cora-LT | CiteSeer-LT | PROTEINS | PTC_MR | MUTAG |
|---|---|---|---|---|---|---|---|
| GCN | vanilla | 2.436 | 2.154 | 2.129 | 12.798 | 4.295 | 2.989 |
| | re-weight | 2.330 | 2.282 | 2.150 | 12.903 | 4.410 | 3.125 |
| | re(/over)-sample | 3.241 | 2.860 | 2.794 | 15.996 | 5.734 | 4.022 |
| | GraphCL | 3.747 | 3.412 | 3.399 | 14.981 | 5.049 | 3.215 |
| | GraphDec | 2.243 | 1.995 | 1.952 | 10.614 | 4.212 | 2.090 |

According to the results, our GraphDec encounters less computation cost than prior methods. The following explains why augmentation doubles the input graph without increasing overall computation costs: (i) The augmentations we adopt (e.g, node dropping and edge dropping) reduce the size of input graphs (i.e., node number decreases 25%, edge number decreases 25-35%); (ii) During each epoch, our GraphDec prunes datasets so that approximately only 50% of the training data is used. (iii) GraphDec prunes the model weights, resulting in a lighter model requiring less computational resources. (iv) Despite the fact that augmentation doubles the number of input graphs, the additional new views only consume forward computational resources without requiring a backward or weight update step, thereby only marginally increases the computation.

