# OpenReview forum: "Diving into Unified Data-Model Sparsity for Class-Imbalanced Graph Representation Learning"
_ICLR.cc/2023/Conference — Submitted to ICLR 2023_

### Official Review · Reviewer_ta4g · 2022-10-24

**Confidence:** 4
**Correctness:** 1
**Technical Novelty And Significance:** 2
**Empirical Novelty And Significance:** 2
**Recommendation:** 3

**Clarity, Quality, Novelty And Reproducibility:**

The problem seems to be new and interesting, but the writing and the organization of the paper is hard for readers to get the idea that the authors try to convey. There are unclarified points, concepts and typos. Here are some examples:

- In Theorem 1, what’s $L^i_{train}$?
- In Theorem 1, why the validation loss in involved?
- In Theorem 1, what’s $\sigma_T$?
- In Theorem 1, what’s the point to upper bound a minimum number?
- Eq. 4: the input is $x^{(t)}$?
- In Eq. 4-5, what’s $\theta_{pruned}$? Is it irrelevant with $t$?


**Details Of Ethics Concerns:**

None.

**Strength And Weaknesses:**

## Strength
(+) The authors provide plentiful empirical results to show the effectiveness of GraphDec.

## Weakness
(-) Motivation is unclear.
- Does the imbalance class problem really exist when massive training data is available?
- Why do the authors choose self-supervised graph contrastive learning? If the model is learned in a self-supervised fashion, does it suffer from class-imbalance?

(-) Key theoretical results seem to be irrelevant with GNNs, graph contrastive learning. Furthermore, I can not see the connections between the proposed methods and the theorem.

(-) Moreover, class-imbalance issue seems not to be considered in method design, making the readers confusing about the improvements in experiments.

(-) The main method seems not to consider any characteristics of graphs and GNNs, but a bag of tricks when applying Data Diet to graph contrastive learning.

(-) The main method also introduces additional computation and memory overheads, which makes readers confusing about its advantages compared to existing graph contrastive learning, sparsity and class-imbalanced learning methods. Besides, the authors did not provide any efficiency checks such as complexity or the running time cost in experiments.

(-) The authors did not compare other pruning methods and the state-of-the-art graph contrastive learning methods. The node classification experiments did not include graph contrastive learning methods either.

(-) There are too many hyperparameters are introduced, but the authors did not test the sensitivity to these hyperparameters.


**Summary Of The Paper:**

This paper argues that current Graph Neural Networks (GNNs) suffer from high time cost when trained on massive data, and class-imbalance issue. The authors then propose GraphDec to address the problems by pruning both GNN and the training data.

**Summary Of The Review:**

I vote for a reject. As pointed out in Weakness, the motivation of the paper is unclear, the proposed methods along with the theory seems to be irrelevant to the problems (the massiveness and class-imbalance in graphs) that the authors try to address, making the main methodology is full of tricks and the paper too technical.

---

> ### Author Response · Authors · 2022-11-18
> **Response to Reviewer ta4g (5/5)**
>
> **Q7**: There are too many hyperparameters are introduced, but the authors did not test the sensitivity to these hyperparameters.
>
> **A7**: Thank you for pointing out this concern. We respectively argue that our GraphDec does not introduce many hyperparameters that need to be manually tuned. The misunderstanding may come from that there are some notations (but not hyper-parameters) used to describe models or datasets. For example, in our latest submission:
>
> ${\mathcal{G}}_{F}$ denotes the graph full dataset;
>
> ${\theta}^{*}$ is the optimal weight parameters of a GNN model;
>
> ${G_{i}\in{\mathcal{G}}_{F}}$ denotes a graph training sample in the full dataset;
>
> $M$ in our first manuscript represents the total size of the dataset  (i.e., $M = |{\mathcal{G}_{F}}|$).
>
> The above notations do not require us to manually set their value.
> In addition, we **do not need to define the dataset's pruning rate** because we employ cosine annealing to gradually reduce the dataset size from 100\% to 0\% throughout the training procedure.
> In the proposed GraphDec, we mainly need to select the pruning rate $\beta$ of the dynamic GNN model. For the model pruning rate, the target pruning ratio for the model is set to 75\%. We use cosine annealing to gradually prune 75\% weights and this setting contributes to improvements among different tasks.
> We additionally test the sensitivity of $\beta$ on the CiteSeer-LT dataset as below.
>
> |Model Target Pruning Ratio $\beta$ (\%)|	0 (full model)| 25 | 50 | 75 (our default setting)| 90 |
> |:-------:|:----------:|:---------:|:---------:|:---------:|:---------:|
> |GraphDec (ours)|77.54	|77.30	|78.02	|78.29	|74.55	|
>
> **Q8**: The writing and the organization of the paper is hard for readers to get the idea that the authors try to convey. There are unclarified points, concepts and typos.
>
> **A8**: Thank you for pointing out the writing problems. According to your suggestions, we have significantly revised the paper. In particular, we have rewritten the method section and revised the abstract and introduction sections for a much better presentation with clear notations. Also, we have corrected typos throughout the paper. All changes are highlighted in red in the revised paper.

---

> ### Author Response · Authors · 2022-11-18
> **Response to Reviewer ta4g (4/5)**
>
> **Q5**: The main method also introduces additional computation and memory overheads, which makes readers confusing about its advantages compared to existing graph contrastive learning, sparsity and class-imbalanced learning methods. Besides, the authors did not provide any efficiency checks.
>
> **A5**: We have provided detailed discussions about the computational cost and memory overheads in Appendix D of our first submission. Specifically, to evaluate the proposed GraphDec's computational cost on a wider range of datasets, we add new results that include three different class-imbalanced node classification datasets (PubMed-LT, Cora-LT, CiteSeer-LT), three different class-imbalanced graph classification datasets (MUTAG, PROTEINS, PTC\_MR), and four baselines (vanilla GCN, re-weight, re(/over)-sample, GraphCL). We run 200 epochs for each method to measure their computational time (second) for training on NVIDIA GeForce RTX 3090 GPU. The results are shown as follows.
>
> |Model|	Method|	PubMed-LT|	Cora-LT|	CiteSeer-LT|	PROTEINS|	PTC\_MR|	MUTAG|
> |:-------:|:----------:|:---------:|:---------:|-------:|:----------:|:---------:|:---------:|
> |GCN|	vanilla|	2.436|	2.154|	2.129|	12.798|	4.295|	2.989|
> |GCN|	re-weight|	2.330|	2.282|	2.150|	12.903|	4.410|	3.125|
> |GCN|	re(/over)-sample|	3.241|	2.860|	2.794|	15.996|	5.734|	4.022|
> |GCN|	GraphCL|	3.747|	3.412|	3.399|	14.981|	5.049|	3.215|
> |GCN|	GraphDec|	2.243|	1.995|	1.952|	10.614|	4.212|	2.090|
>
> The above result demonstrates that our proposed GraphDec has less computation cost than the prior methods. Here, we further explain why GraphDec does not increase the overall computation or memory costs: (i) Our GraphDec prunes the GNN model weights, resulting in a lighter model on computation and memory overheads during training; (ii) In each epoch, our GraphDec prunes dataset so that approximately only 50\% of the training data is used; (iii) The augmentations (e.g., node dropping and edge dropping) reduce the size of input graphs (i.e., node number decreases 25\%, edge number decreases 25-35\%); (iv) Despite the fact that the augmentation doubles the number of the input graphs, additional new views only consume computational resources in the forward pass, but not in the gradient back-propagation pass, thereby only marginally increases the computation cost.
>
> It is worthwhile to mention that the main challenges we want to address are **class imbalance and massive data usage** in graph data. **Avoiding massive computation and memory costs are additional benefits derived from the demand reduction of massive data usage by our model, as supported by the above results.**
>
> **Q6**: The authors did not compare other pruning methods and the state-of-the-art graph contrastive learning methods. The node classification experiments did not include graph contrastive learning methods either.
>
> **A6**: We clarify that the topmost target of our work is improving class-imbalance graph representation learning. However, many other pruning methods and graph contrastive learning methods are not aimed at class-imbalance graph representation learning and are orthogonal to our approaches and can easily be plugged in for achieving dynamic sparsity in models and data. Thus they do not affect our conclusions and contributions. Regardless, we still conduct additional experiments to further address your concern. We apply a new pruning method (Synflow [7]) and two SOTA graph contrastive learning methods (BGRL [8], GRACE [9]) on our class-imbalance node classification datasets. The results show that our dynamic sparse graph contrastive learning model is better than these methods.
>
> |Model (bAcc.)| bAcc.    |bAcc.   |bAcc.    |F1-ma. |
> |:-----------:|:------------:|:------------:|:-----------:|:-----------:|
> |SynFlow       |60.62  |46.23  |56.75   |62.97 |
> |GRACE          |63.95  |50.87 |63.22  |64.52 |
> |BGRL          |64.95 |50.83  |62.21  |63.15 |
> |**GraphDec (ours)** |**73.94**    |**61.56**   |**76.05**      |**92.16** |
>
> [7] Pruning neural networks without any data by iteratively conserving synaptic flow, NeurIPS 2020.
>
> [8] Large-Scale Representation Learning on Graphs via Bootstrapping, ICLR 2022.
>
> [9] Deep Graph Contrastive Representation Learning, ICML Workshop on Graph Representation Learning and Beyond, 2020.

---

> ### Author Response · Authors · 2022-11-18
> **Response to Reviewer ta4g (3/5)**
>
> **Q3**: Moreover, class-imbalance issue seems not to be considered in method design, making the readers confusing about the improvements in experiments.
>
> **A3**: Firstly, we would like to emphasize that our method is designed with the data dynamic sparsity and model dynamic sparsity, both of which are aimed to tackle the class-imbalance issue:
> (a) for data dynamic sparsity, our GraphDec selects a subset to learn more balanced representations, which puts more attention on sampling minority samples, as shown in Figure 5, Appendix D;
> (b) for model dynamic sparsity, our GraphDec uses dynamic sparse graph contrastive learning with the pruned GNN to obtain larger gradients on minority samples, which adaptively assigns higher weights on parameter updates exerted by the minority samples, as shown in Figure 5, Appendix D. Also, a similar observation that pruned neural networks are more sensitive to minority samples is made in the prior work [5].
>
> To verify the effectiveness of considering both data and model dynamic sparsity in addressing the class-imbalance problem, we conduct ablation experiments in Table 3, Section 4.4. We evaluate the performance of GraphDec without training on the sparse dataset (SS), or without training a sparse GNN (SG). The quantitative results show that the performance on class-imbalance graphs significantly dropped. It confirms that considering data and model dynamic sparsity can indeed enhance the performance in class-imbalance graph learning.
>
> [5] Self-Damaging Contrastive Learning, ICML 2021.
>
> **Q4**: The main method seems not to consider any characteristics of graphs and GNNs, but a bag of tricks when applying Data Diet to graph contrastive learning.
>
> **A4**: Thanks for pointing out your concern. However, we respectively emphasize that our method is not a simple combination of different tricks.
> Instead, we propose a novel class-imbalanced graph representation learning framework optimized by a new philosophy,
> aiming to handle the class-imbalanced problem commonly existing in graph learning tasks, therefore we believe our proposed method is meaningful and can be utilized to solve real-world problems in future graph learning research.
>
> More importantly, the main technical contribution of this paper is to discover the dynamic sparsity in the graph (i.e., data level) and GNNs (i.e., model level), and then design a comprehensive framework based on sparsity properties of graph data and GNNs.
> Specifically, our data pruning strategy is definitely not an application of existing data diet methods, because our proposed methods are designed to prune the dataset in a dynamic manner.
> The previous data diet work [6] will find the most influential samples in the initial period, then the subset will be fixed and static for further training.
> However, as we pointed out in our paper, the pruned subsets found by the static data pruning strategy are usually less optimal in the view of the whole training process, because the most influential samples (with the largest gradients) of the full datasets will be changing during training. This conclusion is also empirically proved by the visualization results in Section 4.5 of our paper.
> Meanwhile, we also extend the dynamic sparsity to the model level to dynamically prune the GNN parameters, which further improves the effectiveness of finding the dynamic and most influential samples.
> Experimental results in our paper also prove our model pruning strategy is more effective than the previous model pruning methods.
>
> To the best of our knowledge, this is the first work to study the **dynamic sparsity** in both graph data and the GNN model, we believe our work is novel and can provide an insightful reference for future research on this problem.
>
> [6] Lottery Tickets on a Data Diet: Finding Initializations with Sparse Trainable Networks, NeurIPS 2022.

---

> ### Author Response · Authors · 2022-11-18
> **Response to Reviewer ta4g (2/5)**
>
> **Q1**: Motivation is unclear. (1) Does the imbalance class problem really exist when massive training data is available? (2) Why do the authors choose self-supervised graph contrastive learning? If the model is learned in a self-supervised fashion, does it suffer from class-imbalance?
>
> **A1**:
> (1) Yes, the class-imbalance problem still exists even with massive training data. For example, previous work [1] has demonstrated that it is more challenging to train a deep neural network to well-predict all of the categories of the long-tailed version of the ImageNet dataset than the standard ImageNet dataset.
> To be more specific, the standard ImageNet dataset contains 115.8K images from 1000 categories, with maximally 1280 images per class and minimally 5 images per class. Similar conclusions are also established on other graph datasets according to prior related works [2, 3].
>
> [1] Decoupling Representation and Classifier for Long-Tailed Recognition, ICLR 2020.
>
> [2] GraphENS: Neighbor-Aware Ego Network Synthesis for Class-Imbalanced Node Classification, ICLR 2022.
>
> [3] Imbalanced Graph Classification via Graph-of-Graph Neural Networks, CIKM 2022.
>
> (2) We choose contrastive learning because contrastive self-supervised learning can be modified to dynamically prune one of its branch backbones so as to improve its recognizability on the minority samples in a class-imbalanced dataset, thereby assigning greater values to the minority samples without the supervision of the labels. This observation is also examined in our experiment (i.e., Figure 5 of Appendix D); Also, self-supervised representations are more robust to the class imbalance problem than supervised representations according to the prior related work [4, 5]. So self-supervised learning helps the model to less suffer from class-imbalance problems.
> However, existing self-supervised learning methods cannot fully tackle the class-imbalance problem. According to Table 1 in Experiment 4.2, compared with vanilla GIN, although GraphCL (one representative self-supervised learning method) can alleviate class imbalance, our GraphDec framework can further outperform GraphCL.
>
> [4] Self-supervised Learning is More Robust to Dataset Imbalance, NeurIPS 2021 Workshop on Distribution Shifts.
>
> [5] Self-Damaging Contrastive Learning, ICML 2021.
>
> To make the motivation presentation clearer, we have revised the abstract and introduction sections in the paper (highlighted in red).
>
> **Q2**: Key theoretical results seem to be irrelevant with GNNs, graph contrastive learning. Furthermore, I can not see the connections between the proposed methods and the theorem.
>
> **A2**: Thanks for pointing out your concern. However, we respectively emphasize that the proposed theorem is strongly connected with our proposed framework. First, the contribution of our work is to solve the class-imbalanced issue currently occurring in the graph learning task instead of proposing a new GNN or contrastive learning model.
> Although many existing graph learning methods can achieve extraordinary performance on the class-balanced graph data, those methods can still suffer from the class-imbalanced setting in the graph, including the graph self-supervised learning (SSL) methods, according to our experiments.
>
> Therefore, we focus on proposing a new optimization philosophy for graph learning so that we can use it in a plug-and-play manner to help the existing graph learning models to better adapt to the class-imbalanced setting.
> Our framework is designed to alleviate such class-imbalanced issues with the data pruning mechanism which is based on the (dynamic) **data sparsity** property.
> Specifically, the data sparsity property indicates that the gradients used to update model parameters in the training phase are mostly contributed by a subset of data samples.
>
> Moreover, our experimental results in Appendix D demonstrate that those "significant" (contributes more to the gradients) data samples tend to be highly overlapped with the minority class samples.
> Theorem 1 theoretically justifies the data sparsity property by proving the gradients of a sparse subset can well approximate that of the full dataset, thus achieving similar effects.
>
> **Our proposed GraphDec is thus inspired by the theorem and the observations mentioned above to leverage the data sparsity property to prune the dataset and identify those minority-class samples**, thereby achieving better performances in the class-imbalanced setting.
> During the training, we calculate and rank the gradient of each graph sample in the current training set, then selects the Top-K graph samples as the training set in the next epoch.
> The pruned subset not only can approximate gradient magnitudes of the full dataset but also provide a more class-balanced class distribution to better learn the minority class.
> So, we believe that the theoretical analysis demonstrates the rationality of our design and closely relates to our proposed framework.

---

> ### Author Response · Authors · 2022-11-18
> **Response to Reviewer ta4g (1/5)**
>
> Thank you very much for your valuable comments. We have significantly revised our paper based on your informative feedback and reply to your comments as follows. If our responses resolve your concerns, we genuinely hope that you can increase your score.

---

> > ### Comment · Reviewer_ta4g · 2022-11-21
> > **Thank you. But my attitude remains unchanged.**
> >
> > Thanks for the detailed explanations from the authors.  Unfortunately, my core concern has not been adequately addressed.
> > My core concern is that the whole paper is quite loosely organized. **There is no strong logical connection between the motivation, the thereon, and the method.** Here I only list some examples.
> >
> >  - **The motivation**: I agree that class-imbalance problem still exists even with massive training data.  But why contrastive learning can address this class-imbalance problem is unclear, even with the additional explanation.
> >  >   We choose contrastive learning because contrastive self-supervised learning can be modified to dynamically prune one of its branch backbones so as to improve its recognizability on the minority samples in a class-imbalanced dataset, thereby assigning greater values to the minority samples without the supervision of the labels.
> >
> >  This statement is not rigorous.  For example, I do not think that only contrastive self-supervised learning can “dynamically prune one of its branch backbones so as to improve its recognizability on the minority samples”.
> >  At most, this statement can be viewed as an empirical observation. However, Figure 5 of Appendix D is not enough for the empirical justification for this statement.  Overall, I can't see the necessity of studying self-supervised methods to solve the class-imbalance problem here, especially for the graph data.
> >
> >  - **The theorem**: In my opinion, the theoretical results still do not necessarily apply to SSL GNNs nor consider graph properties. Especially, the theorem has similar conclusions as Data Diet (Paul et al., 2021), hence the so-called “new philosophy” is not new. Furthermore, to obtain the theoretical result, many assumptions are made on the model's parameters and loss function. These assumptions make the theoretical contribution marginal and there is a significant gap between the theoretical findings and real scenarios. Meanwhile, I've checked the proof details of this theorem. It is totally irrelevant to graphs or GNNs. Since this paper aims to solve the class-imbalance problem in graphs, not the general data, the connection between the theoretical part and the other part is weak.
> >
> >  - **The method**: The proposed method is more like a general framework, which combines the model sparsification and sampling boosting technique. This combination is trivial for dealing with graph data since it does not consider any graph properties and only treats GNN as a feature encoder. Furthermore, neither solid theoretical results nor extensive empirical evidence can justify that the proposed method can address the class-imbalance problem.
> >
> > In summary, this paper claims that it can solve the class-imbalance problems in the training of GNNs. However, such a claim is not well supported by the theoretical results and model design, which is important for the research community. **The novelty does not lie in the combination of the complex techniques but exists behind the motivation that we make a such combination in a rational way and why it solves the problem exactly.**
> >
> > Therefore, I still hold a strong negative attitude toward the acceptance of this paper.

---

> > > ### Author Response · Authors · 2022-11-22
> > > **Additional Response to Reviewer ta4g**
> > >
> > > Thanks for your reply. We hope our significantly revised paper can satisfy your suggestions on strengthening logical connections. Also, we point-wisely respond to your concerns as below:
> > >
> > > (1) Motivation: Again, we confirm that dynamic sparse graph contrastive learning is a reasonable and natural choice to identify minority graph samples.
> > > First, prior works [1, 2] have demonstrated that contrastive learning can enable neural networks to map class-imbalanced data into a more balanced feature space than supervised learning.
> > > Second, earlier works [3, 4] show that a supervised pruned deep model is more vulnerable to long-tail samples. A self-supervised pruned deep model, on the contrary, can automatically detect the minority samples so that the learning on them can be strengthened.
> > > We believe that the existing conclusions mentioned above can help you understand our motivation for using sparse graph contrastive learning on imbalance data. We list all the references below, please kindly check.
> > >
> > > [1] Rethinking the Value of Labels for Improving Class-Imbalanced Learning, NeurIPS 2020.
> > >
> > > [2] Exploring balanced feature spaces for representation learning, ICLR 2021.
> > >
> > > [3] What Do Compressed Deep Neural Networks Forget?
> > >
> > > [4] Self-Damaging Contrastive Learning, ICML 2021.
> > >
> > > (2) The theorem: First, as mentioned in the paper, Theorem 1 formally defines the data sparsity property that commonly existed in many datasets and theoretically proves the selected informative (high-quality) subset can be utilized to approximate the gradients of the full dataset, thereby train a deep model with comparable performance.
> > > To the best of our knowledge, it is the first attempt to use the theoretical conclusions to design graph data-sparsity mechanism and apply the theorem on class-imbalance graph tasks.
> > > Therefore, it is logical to claim that we are motivated by Theorem 1 to design the dynamic selection of class-balanced graph data subset in GraphDec.
> > > Second, it is very common to have scenarios (e.g., [7]) in which we need to add some reasonable assumptions during the implementation of experiments, especially when solving some more practical problems like this work.
> > >
> > > (3) Method: Firstly, we believe our GraphDec brings specific improvements to class-imbalance graph representation learning. For example, on node classification tasks, since our framework removes many uninformative nodes, the final graph structure is largely simplified and then we reduce overall computational costs commonly brought by graph structures. Secondly, the generality of our work implies that the design is not limited to graph tasks but also has helpful potential in other modalities (such as images or texts). Although transferring GraphDec from graph data to Euclidean data will lose the benefits of simplifying the structure of data, we believe the unified dynamic data-model sparsity method might still train a high-quality deep model with less data usage in real-world scenarios (such as noisy data or class imbalance).
> > >
> > > To sum up, your core concern is that our proposed framework is overly general and not specific to graphs.
> > > So, we want to present our motivation for the research on graphs here.
> > > Despite there exist differences between graph data and other data like images and text, graphs can still be considered a common data form and share many properties with the so-called "general data".
> > > We believe it would be more helpful to raise the impact of the graph learning community if we can propose methods to solve some general and practical problems instead of being obsessed with graph- and GNN-specific properties and thus limiting the research scope.
> > > In fact, graph learning research is just developing in this direction, and many recent works about graph learning can be used to prove our point. Here, we only list a few of them for your reference.
> > > [5] provides an ELBO bound, and primarily focuses on the disentangling problem without considering too many exact graph properties.
> > > [6] studies the feature-level decorrelation to learn a better graph representation without relying on negative sampling in contrastive learning.
> > > We think the methods proposed by the these papers can be considered as the so-called "general framework" and they are evaluated on the graph data without considering too many graph-specific properties.
> > > However, the performance is obviously improved and we do not doubt that they can be inspiring for further research of graphs and other fields.
> > > We do not run experiments to evaluate our proposed GraphDec in other "general data" to see if it is as effective.
> > > However, we will be glad to know if our GraphDec is a "general framework", and we do not think this should be taken as a weakness of our work. Thanks!
> > >
> > > [5] Disentangled Contrastive Learning on Graphs, NeurIPS 2021.
> > >
> > > [6] From Canonical Correlation Analysis to Self-supervised Graph Neural Networks, NeurIPS 2021.
> > >
> > > [7] InstaHide: Instance-hiding Schemes for Private Distributed Learning, ICML 2020.

---

> > > > ### Comment · Reviewer_ta4g · 2022-11-29
> > > > **Thank you for your follow-up response.**
> > > >
> > > > Since the ICLR is a open peer-reviewed community, it's good to make the in-depth discussions.
> > > > After reading the follow-up response, let's narrow down our discussion point.
> > > > 1. For the motivation part, I finally understand what your motivation comes from. I haven't   checked the reference carefully but for now, let's agree that it is reasonable to employ the contrastive and self-supervised learning to address class-imbalanced problem.
> > > >
> > > > 2. For the theorem part, first point, if this paper is focus on solve the problem in graphs while this theorem is totally unrelated to graphs, i.e., no graph properties used during training and in the final main results, I would treat it as a nice ``math barrier'', which is used to decorate the paper, rather than a solid contribution here.  Second point, the more assumption you made in your proof, the less contribution you made in your theorem.  So, still, the theoretical contribution of the paper is weak.
> > > >
> > > > 3. For the method part, if I do not misunderstand (correct me if I'm wrong ), the term "data-model sparsity" you mentioned is the sparsity between the graph samples, not for the node or the graph topology.  The only part may related to graph property in this method is that the graph argumentation in the contrastive learning step. However, I think this part is not very important for this framework. For the general data, we can still take the proper argumentation techniques to generate the two views for the following contrastive learning and gradient ranking.
> > > >
> > > >
> > > >  In one sentence,  **If your framework can deal with the general data, show me the evidence and update your claim.  I would consider to give an acceptance.** For the current state, I still think the acceptance of this paper is very reluctant, even with two high ratings.

---

> > > > > ### Author Response · Authors · 2022-12-01
> > > > > **Additional response to Reviewer ta4g**
> > > > >
> > > > > Thanks for your reply. We are happy to see your recognition of our motivation in using contrastive learning, and we feel excited to discuss more our theorem and method parts. We have also conducted further experiments on your open problem regarding the GraphDec framework's application on general data.
> > > > >
> > > > > For the theorem part, we would like to emphasize that the conclusion of our theorem can motivate us to design a data-sparsity framework for selecting high-quality subsets to solve class-imbalanced problem in general data. In addition, it is also a significant contribution that we are the first to effectively apply this theorem to class-imbalanced graph learning problem.
> > > > >
> > > > > For the method part, we would like to clarify that the data sparsity in the term "data-model sparsity" refers to not only sparsity between the graph samples on graph-level classification tasks, but also sparsity between nodes on node-level classification tasks (since our GraphDec also considers pruning unimportant nodes on node classification datasets such as Cora-ML, which prunes the graph structure and reduces the number of nodes. It satisfies the meaningful consideration of graph properties).
> > > > >
> > > > > For experiments, we further evaluate the application of GraphDec on a general benchmark image dataset: CIFAR-10, where we can compare GraphDec with the related methods including Forget Score [1] and Data Diet (EL2N version and GradNd version) [2]. We use ResNet18 [3] as the backbone and other detailed settings for hyperparameters are consistent with Data Diet.
> > > > > The comparison results (each result refers to the average of multiple independent runs) are displayed in the table below:
> > > > >
> > > > > | Strategy (CIFAR10 + ResNet18)            | Test Accuracy(\%) | Data Sparsity(\%) |
> > > > > |----------------------------------------|---------------|---------------|
> > > > > | No Pruning                             |         95.27       |      100          |
> > > > > | Pruning Randomly                       |          93.38     |       50         |
> > > > > | Forget Score [1] with 200 epochs for selection           |      95.34         |       50         |
> > > > > | Data Diet [2] (EL2N Score with 20 epochs for selection) |         95.21      |      50          |
> > > > > | Data Diet [3] (GradNd Score at initialization)   |          95.16     |      50          |
> > > > > | **Ours (GraphDec)**                        |          **95.66**     |      50          |
> > > > >
> > > > > The above table shows that our GraphDec can outperform the prior methods (we would like to emphasize there is a common consensus in computer vision that achieving a marginal improvement of the image classification task on CIFAR10 is not easy). In addition, the second-best method Forget Score needs an additional subset selection period for 200 epochs, which is not included in the regular training period (200 training epochs) and uses much more epoch numbers than our framework. **Considering both accuracy and training epoch numbers, our GraphDec framework reflects its promising potential on more general data.**
> > > > >
> > > > > **We think the above-updated response and experiments can further support our conclusions in the paper. If our replies have addressed your concerns, we sincerely hope you can increase your evaluation score. Thank you very much.**
> > > > >
> > > > > [1] An empirical study of example forgetting during deep neural network learning, ICLR 2019
> > > > >
> > > > > [2] Deep Learning on a Data Diet: Finding Important Examples Early in Training, NeurIPS 2021
> > > > >
> > > > > [3] Deep Residual Learning for Image Recognition, CVPR 2016

---

> > > > > > ### Comment · Reviewer_ta4g · 2022-12-06
> > > > > > **The final resospone.**
> > > > > >
> > > > > > I appreciate the effort of continuing to refine the paper and making clarifications from the authors. Let me finalize our discussion here.
> > > > > >
> > > > > > 1. I still can't recognize your theoretical contribution, especially for the graph. You can claim that you are "motivated" by the theorem, but unfortunately, your model **is not based on**  the theorem, which greatly reduce the contribution of the theoretical part.
> > > > > >
> > > > > > 2. After carefully checking your paper, I confirm that your method can deal with node classification task (this part is vague in the main text. ). However,  the "sparsity" talked about in this paper is quite different from the "topology sparsity" in many GNN papers. Here, the "data sparsity" refers to the node sample, not the edge sparsity. I hope the author does not confuse these concepts. Essentially,  your method **is not constructed specifically to the graph data and GNN model**.
> > > > > >
> > > > > > 3. The new experimental results look promising.  I think this method is more suitable described on the general data.
> > > > > >
> > > > > > In a summary, if you can revise the paper by :
> > > > > > 1. removing the claims about graphs / GNN, or at least, do not describe your motivation with graph and do not use the name such as GraphDec,
> > > > > > 2. adding the above experiments to the main text, not just the appendix.
> > > > > >
> > > > > > I will consider to accept this paper.

---

> > > > > > > ### Author Response · Authors · 2022-12-07
> > > > > > > **Revision per Reviewer ta4g's request**
> > > > > > >
> > > > > > > Thanks for your insightful suggestions and the recognition of our work. We have revised the paper per your comments. At this moment, we are not allowed to update the PDF but we promise to include the following revisions in the final version once accepted. Yes, our proposed method is a general framework for various data such as graphs and images. In this work, we primarily test the effectiveness of our proposed method on the non-Euclidean data (graphs), where the class-imbalance issue is particularly accentuated. Additional experiments on image data are also conducted to demonstrate the generalization ability of our method on other data.
> > > > > > >
> > > > > > > 1. *removing the claims about graphs / GNN, or at least, do not describe your motivation with graph and do not use the name such as GraphDec.*
> > > > > > >
> > > > > > > We have revised the following parts of our paper per the above request.
> > > > > > >
> > > > > > > (1) We re-name our framework as **DataDec**, replace "Graph Decantation" with "Data Decantation", and modify related module names to show our framework's generality over data across multiple domains.
> > > > > > >
> > > > > > > (2) In the abstract, rather than specifically stating that our method is designed for graphs, we generally point out that the problem is prevalent in all kinds of data and accentuated on the non-Euclidean data (graph) due to its irregularity, and use graph data as a typical example - "This problem is particularly accentuated when deep learning models such as Graph Neural Networks (GNNs) are trained upon non-Euclidean data (e.g., graphs)".
> > > > > > >
> > > > > > > At the end of the abstract, we also stress the validation of our framework on not only graph data, but also on image data CIFAR-10
> > > > > > > - "In our experiments, the effectiveness of
> > > > > > > DataDec is primarily evaluated on multiple graph benchmark datasets. We also
> > > > > > > conduct experiments on image data, i.e., CIFAR-10, to prove the generalization
> > > > > > > ability of DataDec".
> > > > > > >
> > > > > > > (3) In the introduction section, rather than straightly pointing out that the framework is specifically designed for graphs, we state that our focus is to propose a general method to solve the problem and primarily test its effectiveness with non-Euclidean data (graph). Besides, additional experiments on image data (CIFAR-10) are also conducted to evaluate its generalization ability.
> > > > > > > - "Non-Euclidean structured data extensively exists in a wide range of application domains, and a considerate amount of them are naturally abstracted into graphs (e.g., social networks, biochemical
> > > > > > > molecules, knowledge graphs)", "We conduct comprehensive experiments, primarily on multiple class-imbalance graph benchmark datasets for the graph and node classification tasks to demonstrate DataDec's effectiveness, and CIFAR-10 for the image classification to verify its generality".
> > > > > > >
> > > > > > > (4) In the methodology section, we replace all the biased notations indicating that the dataset is restrained to graphs. We also refine our statement by replacing phrases like "graph dataset" with "dataset", and "graph subset" with "data subset", except when we are using graphs for exemplifying.
> > > > > > >
> > > > > > > (5) In the experiment section, we state that the effectiveness of our framework is verified by the graph benchmarks, and we further use CIFAR-10 to validate its generality - "we conduct extensive experiments to validate the effectiveness and generalization ability of our proposed model for both the graph and node classification tasks on class-imbalanced graph datasets, as well as the image classification task on CIFAR-10".
> > > > > > >
> > > > > > > 2. *adding the above experiments to the main text, not just the appendix*
> > > > > > >
> > > > > > > Per the above request, we have added the experiment results on CIFAR-10 comparing our framework with other related methods, as well as the analysis to the experiment section in the main text, to stress the generality of our framework - it is easily adapted to data from other domains.
> > > > > > >
> > > > > > > **Thank you for recognizing the contributions of our work. We hope the above revisions can clear your concern to accept our paper.** Please let us know if you have any more questions.

---

> > > > > ### Author Response · Authors · 2022-12-05
> > > > > **Response reminder**
> > > > >
> > > > > Dear reviewer ta4g,
> > > > >
> > > > > It has been a while since we submitted our last response regarding your further questions and experiments on general data. Since we are approaching the discussion period end, we hope that you can take a look on our response and kindly reconsider your evaluation. Please let us know if you have any more questions. Thank you.

---

### Official Review · Reviewer_9iT6 · 2022-10-24

**Confidence:** 5
**Correctness:** 4
**Technical Novelty And Significance:** 4
**Empirical Novelty And Significance:** 3
**Recommendation:** 8

**Clarity, Quality, Novelty And Reproducibility:**

Core resources (e.g., code, data) are available. The submission contains sufficient details (e.g., proofs, experimental setups) to be useful and clear to other researchers. The overall method is novel and ingenious.


**Strength And Weaknesses:**

Strength:
1. The idea of unifying dynamic informative subset selection and subnetwork selection during training is a first exploration and compelling. In addition, as a rebalancing method in this paper, data and model co-pruning do not require additional learnable weights or human-crafts hyperparameters. It can be used as a plug-and-play tool for other GNN models.

2. The model design is well-driven by an intriguing hypothesis. It theoretically examines the extent to which a subset of the training data can approximate the learning efficiency of the entire dataset. Based on the hypothesis, the methods and experiments also demonstrate the rebalanced subset's gradients can effectively approximate the entire imbalanced dataset.

3. The authors suggest Graph Decantation (GraphDec) as a solution to the difficulties posed by training on huge class-imbalanced graph data. Utilizing the self-pruned contrastive framework's sensitivity on informative samples to gradually refine a more balanced subset is an innovative approach. Consequently, this subset can reduce learning computation and train the model contributing more balanced predictions for classifying graph data.

4. The experiments are sufficient and the improvement of GraphDec is significant. Particularly, the experiments studying the "Evolution of Sparse Subset by Scoring All Samples" are interesting, which supports the paper's motivation and inspires readers.


Weakness:
1. Does the idea of dynamically mining informative samples be applicable to some supervised frameworks? It will be nice to discuss the effect of applying the self-supervised learning (SSL) framework since the motivation for using SSL needs to be strengthened. Also, I think the problem of imbalance classification on the graph is very general among supervised and self-supervised settings. It is better to analyze the generality of GraphDec.

2. When I see the GraphDec sorting neural weights and data samples for pruning, I am expecting the exact computation consumption of this model-data joint optimization can be clearly shown. I am not sure whether the method can actually trim down the computational cost.

**Summary Of The Paper:**

This paper addresses the class-imbalance problem in graph learning. The authors propose Graph Decantation (GraphDec) framework for learning balanced graph representation in a self-supervised manner. The key procedure of GraphDec is to select some informative samples from unbalanced node data or graph data during training based on the gradient norm. The idea is interesting and novel, and the method is generally applicable to either node-level or graph-level class-imbalance situations.

**Summary Of The Review:**

This is a technically sound paper with novel ideas on dynamic graph learning model and data sparsity, and its impact on the community is not restricted to the graph area. I believe that the experimental evaluation results are encouraging and sufficient. I would like to accept this novel and solid work.

---

> ### Author Response · Authors · 2022-11-18
> **Response to Reviewer 9iT6**
>
> We truly appreciate your feedback. Following are clarifications to your questions and concerns.
>
> **Q1**: Does the idea of dynamically mining informative samples be applicable to some supervised frameworks? It will be nice to discuss the effect of applying the self-supervised learning (SSL) framework since the motivation for using SSL needs to be strengthened. Also, I think the problem of imbalance classification on the graph is very general among supervised and self-supervised settings. It is better to analyze the generality of GraphDec.
>
> **A1**: [Does the idea of dynamically mining informative samples be applicable to some supervised frameworks?] Yes, our idea of mining dynamic sparsity of graph dataset are also applicable to supervised learning. We respectively underline our motivation for using contrastive self-supervised learning in GraphDec:
> First, contrastive learning can be modified to dynamically prune one of its branch backbones so as to improve its recognizability on the minority samples in an class-imbalanced dataset, thereby assigning greater values to the minority samples without the supervision of the labels. This observation is also examined in our experiment (i.e., Figure 5 of Appendix D); Second, self-supervised representations are more robust to the class imbalance problem than supervised representations according to the prior related work [1, 2]. Therefore, in this work, we only use the contrastive method in our framework.
>
> [It is better to analyze the generality of GraphDec.] In GraphDec, we first find that dynamic sparsity, a very general characteristic, is existing in both graph data and GNN models. Also, we find that in computer vision tasks, sparsity also has been applied to train DNNs for more robustness.
> Based on this observation and prior work [3] on robustness, we believe our GraphDec has the potential to inspire us or the community to develop more supervised and self-supervised methods to leverage dynamic sparsity for more general graph mining applications not limited to class-imbalance learning (e.g., graph adversarial robustness).
>
> [1] Self-supervised Learning is More Robust to Dataset Imbalance, NeurIPS 2021 Workshop on Distribution Shifts.
>
> [2] Self-Damaging Contrastive Learning, ICML 2021.
>
> [3] Sparsity Winning Twice: Better Robust Generalization from More Efficient Training, ICLR 2022.
>
> **Q2**: When I see the GraphDec sorting neural weights and data samples for pruning, I am expecting the exact computation consumption of this model-data joint optimization can be clearly shown. I am not sure whether the method can actually trim down the computational cost.
>
> **A2**: We have provided more discussions about the computational cost in our first submission's Appendix E. Specifically, To evaluate the proposed GraphDec's computational cost on a wider range of datasets, besides the results in the previous response, we add new results that include three different class-imbalanced node classification datasets (PubMed-LT, Cora-LT, CiteSeer-LT), three different class-imbalanced graph classification datasets (MUTAG, PROTEINS, PTC\_MR), and four baselines (vanilla GCN, re-weight, re(/over)-sample, GraphCL). We run 200 epochs for each method to measure their computational time (second) for training on NVIDIA GeForce RTX 3090 GPU. The results are shown as follows.
>
> |Model|	Method|	PubMed-LT|	Cora-LT|	CiteSeer-LT|	PROTEINS|	PTC\_MR|	MUTAG|
> |:-------:|:----------:|:---------:|:---------:|-------:|:----------:|:---------:|:---------:|
> |GCN|	vanilla|	2.436|	2.154|	2.129|	12.798|	4.295|	2.989|
> |GCN|	re-weight|	2.330|	2.282|	2.150|	12.903|	4.410|	3.125|
> |GCN|	re(/over)-sample|	3.241|	2.860|	2.794|	15.996|	5.734|	4.022|
> |GCN|	GraphCL|	3.747|	3.412|	3.399|	14.981|	5.049|	3.215|
> |GCN|	GraphDec|	2.243|	1.995|	1.952|	10.614|	4.212|	2.090|
>
> The above result demonstrates that our proposed GraphDec has less computation cost than prior methods. Here, we further explain why GraphDec does not increase the overall computation or memory costs: (i) Our GraphDec prunes the GNN model weights, resulting in a lighter model on computation and memory overheads during training; (ii) In each epoch, our GraphDec prunes dataset so that approximately only 50\% of the training data is used; (iii) The augmentations (e.g., node dropping and edge dropping) reduce the size of input graphs (i.e., node number decreases 25\%, edge number decreases 25-35\%); (iv) Despite the fact that augmentation doubles the number of the input graphs, additional new views only consume forward computational resources without requiring a backward step or a weight update step, thereby only marginally increase the computation cost.
>
> It is worth mentioning that the main challenges we present in this paper are **massive data usage and class imbalance** in graph data. **Avoiding massive computation and memory costs are additional benefits derived from the demand reduction of massive data usage by our model, as supported by the above results.**

---

### Official Review · Reviewer_ZWS9 · 2022-10-25

**Confidence:** 5
**Correctness:** 4
**Technical Novelty And Significance:** 3
**Empirical Novelty And Significance:** 3
**Recommendation:** 8

**Clarity, Quality, Novelty And Reproducibility:**

The paper is well-written and the methodology is well-motivated. The designed model is novel and solid. The availability of code and data facilitates reproducibility.

**Strength And Weaknesses:**

+This work is solid and insightful. The proposed framework investigates a new perspective in which sparsification can be utilized in both graph datasets and graph neural networks, which possess many advantages for real-world applications. For example, this work can be leveraged to detect important data points and conserve computational time.

+The authors conduct comprehensive ablation studies to evaluate each model components, which clearly show the contributions of every module and the impact of rebalancing strategies.
The introduced components such as dynamically downsampling important subsets, and pruning contrastive model are empirically proven to contribute distinct positive impacts to the full framework. Despite employing a subset of the data, the proposed method achieves similar or better performance compared to the methods that utilize the entire dataset.

+ The overall structure is based on an intriguing hypothesis with proof. The authors provide a hypothesis to explain why downsample a subset can estimate the entire graph dataset; they infer that the subset with the gradient closest to the gradient of the entire dataset is the most important. They then apply this inspiration to develop GraphDec. In addition, the inspiration provided by this hypothesis is also relevant for tasks in other domains (e.g., computer vision, natural language processing).

Weakness/questions:
-What is the purpose of training the contrastive learning system? Does this method exclude other learning schemes and supervised loss? The paper can be improved by highlighting the unique effect, such as a concise summary of the motivation and empirical ablation experiments.

-Certain technical descriptions lack clarity for readers unfamiliar with coresets. In the experiment section, for instance, the word "prune" is sometimes used in a rather ambiguous manner. In the previous method section, it refers to the network, whereas in the experiment section, prune appears to be the "sampling" step of the dataset. The authors should provide additional information to explain the definitions of prune and sample on data and model or clarify that the terms are used in different sentences.



**Summary Of The Paper:**

The paper aims to solve both graph data imbalance and unnecessary model-level computation burden in a unified framework. Specifically, the authors first examine the challenges from theoretical and empirical perspectives. Then, the authors propose GraphDec, a novel data-model dynamic sparsity framework to address the challenges. Extensive experiments on multiple benchmark datasets demonstrate that GraphDec outperforms state-of-the-art methods.

**Summary Of The Review:**

The paper address two significant challenges of imbalanced graph learning by designing a novel and effective model. The technical contribution is solid. The empirical explanations are interesting, and the experimental results are remarkable.

---

> ### Author Response · Authors · 2022-11-18
> **Response to Reviewer ZWS9**
>
> Thank you very much for your comments. We make clarifications to your questions and concern as follows.
>
> **Q1**: What is the purpose of training the contrastive learning system? Does this method exclude other learning schemes and supervised loss? The paper can be improved by highlighting the unique effect, such as a concise summary of the motivation and empirical ablation experiments.
>
> **A1**: The purpose of using contrastive learning can be illustrated in two perspectives:
> (a) contrastive self-supervised learning can be modified to dynamically prune one of its branch backbones so as to improve its recognizability on the minority samples in an class-imbalanced dataset, thereby assigning greater values to the minority samples without the supervision of the labels. This observation is also examined in our experiment (i.e., Figure 5 of Appendix D);
> (b) self-supervised representations are more robust to the class imbalance problem than supervised representations according to the prior related work [1, 2]. Therefore, in this work, we design a modified contrastive learning framework to solve class-imbalanced graph representation learning problem.
>
> We have revised the abstract and introduction of our paper (highlighted in red) to improve the clarity regarding the motivation for using contrastive learning.
>
> [1] Self-supervised Learning is More Robust to Dataset Imbalance, NeurIPS 2021 Workshop on Distribution Shifts.
>
> [2] Self-Damaging Contrastive Learning, ICML 2021.
>
>
> **Q2**: Certain technical descriptions lack clarity for readers unfamiliar with coresets. In the experiment section, for instance, the word "prune" is sometimes used in a rather ambiguous manner. In the previous method section, it refers to the network, whereas in the experiment section, prune appears to be the "sampling" step of the dataset. The authors should provide additional information to explain the definitions of prune and sample on data and model or clarify that the terms are used in different sentences.
>
> **A2**: Thank you for pointing out the concern. First, we have revised the method section (highlighted in red) to clarify all notations of operations and made sure they are consistent with the experimental section. Second, we want to clarify that the pruning on the dataset denotes that we downsample a subset from the full dataset, and the pruning on GNN denotes that we remove some unimportant neurons from the full GNN model. For better clarity, when describing the operations on datasets, we replace all "pruning" with "sampling"; when describing the slimming operations on GNN models, we consistently use "pruning".

---

### Author Response · Authors · 2022-11-18
**Response to All Reviewers**

We gratefully thank all reviewers for your valuable comments and insightful suggestions. We appreciate the recognition of the sparsification problems (existing in both graph and GNN) GraphDec tackles [by reviewer ZWS9 and 9iT6], the easily-implemented property of GraphDec possesses [by reviewer 9iT6], as well as the agreement on the extensiveness of our experiments in confirming the effectiveness of GraphDec [by reviewer ZWS9, 9iT6 and ta4g].

Our work is firstly inspired by the observation of the dynamic sparsity characteristics in class-imbalanced graphs and GNNs. Based on the use of dynamic sparsity, we design the GraphDec framework, with a series of techniques guiding dynamic sparsity training from both the model and data aspects for more class-balanced representations. Exploiting such characteristics helps to facilitate the models in learning more class-balanced representations that are more general with respect to various downstream tasks.

We also notice that reviewers raised several concerns about the paper, such as the motivation of adopting graph contrastive learning [by reviwer ZWS9 and ta4g], the actual effects in relieving the computational burdens [by reviewer 9iT6 and ta4g], and the GraphDec's relations to graphs and GNNs [by reviewer ta4g]. To address and clarify these concerns, we run extra experiments concerning each of the reviewer's requirements, illustrate and provide our thoughts regarding the posted questions. In addition, to address reviewer ta4g's concerns about the writing and clarity, we significantly revise the paper writing and presentation (especially for the introduction and method sections) to make the paper presentation better.

We again appreciate the valuable comments and suggestions to our work from several perspectives. Please let us know if your concerns are not fully addressed, and we are happy to answer any more questions you may have.

---

> ### Author Response · Authors · 2022-11-25
> **reminder**
>
> Dear reviewers,
>
> Thank you again for reviewing our paper. It has been a while since we submitted our response. We hope we have addressed your comments and your feedback in the discussion period is important to us.

---

### Decision · Program_Chairs · 2023-01-20

**Decision:**

Reject

**Justification For Why Not Higher Score:**

The SAC and AC found serious issues with the key theorem in the paper (Theorem 1) during the discussion phase. It seems to be a rehashing of a standard result about using a stochastic gradient in place of the true gradient, and it is not clear what (if any) specialized properties are leveraged here. Further, several steps in the theorem are incorrect or unclear. For example, the term Err^t includes L_{G_F}, but L_{G_F} does not appear in the proof. Also, the theorem is quite unrelated to graphs, and there are several strong assumptions.

**Justification For Why Not Lower Score:**

NA

**Metareview: Summary, Strengths And Weaknesses:**

The paper first examines the challenges in graph data imbalance and model-level computational burden theoretically and empirically. A GraphDec model has been proposed to address these challenges and evaluated on multiple benchmark datasets. The GraphDec encompasses dynamically choosing a subset of samples in the graph data and also dynamically prune the GNN during the training process.

Solving the class-imbalanced graph learning problem through a dynamic data-model co-sparsity approach is interesting and novel. There are sufficient ablation studies to emphasize the significance of each algorithmic component.

The theorem 1, which is a key pivot for the entire paper seems flawed with several steps of the theorem being unclear and incorrect.





**Summary Of Ac-Reviewer Meeting:**

Reviewers 9iT6 and ZWS9 found the paper to have a clear motivation and novel, with sufficient empirical evidence.

However, the The SAC and AC found serious issues with the key theorem in the paper (Theorem 1) during the discussion phase. It seems to be a rehashing of a standard result about using a stochastic gradient in place of the true gradient, and it is not clear what (if any) specialized properties are leveraged here. Further, several steps in the theorem are incorrect or unclear. For example, the term Err^t includes L_{G_F}, but L_{G_F} does not appear in the proof. Also, the theorem is quite unrelated to graphs, and there are several strong assumptions.